



# Impact of a moderate volcanic eruption on chemistry in the lower stratosphere: balloon-borne observations and model calculations

Gwenaël Berthet[1], Fabrice Jégou[1], Valéry Catoire[1], Gisèle Krysztofiak[1], Jean-Baptiste Renard[1], Adam E. Bourassa[2], Doug A. Degenstein[2], Colette Brogniez[3], Marcel Dorf[4], Sebastian Kreycy[4], Klaus Pfeilsticker[4], Bodo Werner[4], Franck Lefèvre[5], Tjarda J. Roberts[1], Thibaut Lurton[1], Damien Vignelles[1], Nelson Bègue[6], Quentin Bourgeois[7], Daniel Daugeron[1], Michel Chartier[1], Claude Robert[1], Bertrand Gaubicher[1], and Christophe Guimbaud[1]

[1]Laboratoire de Physique et Chimie de l'Environnement et de l'Espace (LPC2E), Université d'Orléans, CNRS UMR7328, Orléans, France
[2]Institute of Space and Atmospheric Studies, University of Saskatchewan, Saskatoon, Canada
[3]Laboratoire d'Optique Atmosphérique, Université Lille 1 Sciences et Technologies, CNRS UMR8518, Villeneuve d'Ascq, France
[4]Institute of Environmental Physics, University of Heidelberg, Heidelberg, Germany
[5]Laboratoire Atmosphères Milieux Observations Spatiales, UPMC, Université Paris 06, Université Versailles Saint Quentin, CNRS UMR8190, LATMOS-IPSL, Paris, France
[6]Laboratoire de l'Atmosphère et des Cyclones, UMR8105 CNRS, Université de la Réunion, France
[7]Department of Meteorology and Bolin Centre for Climate Research, Stockholm University, Stockholm, Sweden

**Abstract.**

The major volcanic eruption of Mount Pinatubo in 1991 has been shown to have significant effects on stratospheric chemistry and ozone depletion even at mid-latitudes. Since then, only "moderate" but recurrent volcanic eruptions have modulated the stratospheric aerosol loading such as the eruption of the mid-latitude Sarychev volcano which injected 0.9 Tg of sulfur dioxide (about 20 times less than Pinatubo) in June 2009. In this study, we investigate the chemical impacts of the enhanced liquid sulfate aerosol loading resulting from this moderate eruption using data from a balloon campaign conducted in northern Sweden (Kiruna-Esrange, 67.5°N, 21.0°E) in August-September 2009. Balloon-borne observations of $NO_2$, $HNO_3$ and BrO from infrared and UV-visible spectrometers are compared with the outputs of a three-dimensional (3D) Chemistry-Transport Model (CTM). It is shown that differences between observations and model outputs are not due to transport calculation issues but rather reflect the chemical impact of the volcanic plume below 19 km in altitude. Good measurement-model agreement is obtained when the CTM is driven by volcanic aerosol loadings derived from in situ or space-borne data. As a result of enhanced $N_2O_5$ hydrolysis in the Sarychev volcanic aerosol conditions, the model calculates reductions of ~45% and increases of ~11% in $NO_2$ and $HNO_3$ amounts respectively over the summer 2009 period. The decrease in $NO_x$ abundances is limited due to the expected saturation effect for high aerosol loadings. The links between the various chemical catalytic cycles involving chlorine, bromine, nitrogen and $HO_x$ compounds in the lower stratosphere are discussed. The increased BrO amounts (~22%) compare rather well with the balloon-borne observations when volcanic aerosol levels are accounted for in the CTM and appear to be mainly controlled by the coupling with nitrogen chemistry rather than by enhanced $BrONO_2$ hydrolysis. Simulated effects of the Sarychev eruption on chlorine activation and partitioning are very limited in the high temperature conditions in the stratosphere at the period considered, inhibiting the effect of $ClONO_2$ hydrolysis. As a consequence, the simulated ozone loss due to the Sarychev aerosols is low with a reduction of 1.1% of the ozone budget at 16.5 km. Some comparisons with the reported Pinatubo chemical impacts are also provided and overall the Sarychev aerosols have led to less chemical effects than the Pinatubo event.



# 1. Introduction

In the stratosphere, the photo-oxidation of $N_2O$ is the main source of the total nitrogen species ($NO_y$). About 97% of the stratospheric $NO_y$ budget can be explained by the $NO$, $NO_2$, $HNO_3$, $ClONO_2$, and $N_2O_5$ compounds and the partitioning between reactive and reservoir nitrogen species is an important issue in stratospheric ozone chemistry (e.g. Wetzel et al., 2002; Brohede et al., 2008). Nitrogen oxides ($NO_x = NO + NO_2$) are major catalysts responsible for significant ozone destruction in the middle stratosphere. In the gas phase, $NO_x$ interacts with the hydrogen and halogen species in catalytic cycles affecting ozone loss rates in the lower stratosphere (e.g. Portmann et al., 1999; Salawitch et al., 2005). Therefore $NO_x$ can also buffer the ozone destruction by halogenated compounds through the formation reaction of $ClONO_2$ and $BrONO_2$ (e.g. Rivière et al., 2004). The $HNO_3$ reservoir is formed from $NO_x$ indirectly via the hydrolysis of $N_2O_5$ on liquid sulfate aerosols:

$$N_2O_5 + H_2O_{(aq)} \rightarrow 2\ HNO_3 \quad (1)$$

It has been shown that models need to include reaction (1) to better reproduce observations of $NO_y$ partitioning at mid-latitude for background aerosol conditions (i.e. in volcanically quiescent periods) in the lower stratosphere (Rodriguez et al., 1991; Granier and Brasseur, 1992; Fahey et al., 1993; Webster et al., 1994; Salawitch et al., 1994b; Sen et al., 1998). This reaction tends to decrease $NO_x$ amounts and reduces the ozone loss efficiency associated with the $NO_x$ catalytic cycle as the less reactive nitrogen reservoir $HNO_3$ is formed (e.g. Rodriguez et al., 1991; Weisenstein, 1991; McElroy et al., 1992). Reaction (1) is fairly insensitive to temperature and has the potential to greatly reduce reactive nitrogen globally, even under background aerosol conditions.

The hydrolysis of $ClONO_2$ can be expressed by:

$$ClONO_2 + H_2O_{(aq)} \rightarrow\ HNO_3 + HOCl \quad (2)$$

It results in additional formation of $HNO_3$ on sulfate aerosols and to the formation of reactive chlorine in the sunlight where $HOCl$ is rapidly photolyzed releasing $Cl$ radicals (e.g. Hofmann and Solomon, 1989; Prather, 1992; McElroy et al., 1992). This heterogeneous reaction is highly dependent on the water content in the aerosols and has been shown to be of considerable importance in determining the abundance of active chlorine available to destroy ozone under some conditions, i.e. for temperatures typically below 210-215 K and where $HNO_3$ photolysis rates are slow (typically in winter at high latitudes), (e.g. Hanson et al., 1994; Tie et al., 1994; Borrmann et al., 1997). However, for higher temperatures the $ClONO_2$ hydrolysis is not expected to be significant enough to compete with reaction (1) on the $NO_y$ partitioning under these conditions (Fahey et al., 1993; Cox et al., 1994; Sen et al., 1998). Also, the reaction,

$$ClONO_2 + HCl_{(aq)} \rightarrow\ HNO_3 + Cl_2 \quad (3)$$

of $ClONO_2$ with dissolved $HCl$ in sulfuric acid droplets have negligible effects on chlorine activation at such temperatures (Hanson et al., 1994; Borrmann et al., 1997).
Some works also suggest that the hydrolysis of $BrONO_2$,

$$BrONO_2 + H_2O_{(aq)} \rightarrow\ HNO_3 + HOBr \quad (4)$$

on background sulfate aerosols also plays a significant role in ozone depletion in the lower stratosphere with rates almost independent of temperature making this reaction efficient at all latitudes and for all seasons (Hanson and Ravishankara, 1995; Hanson et al., 1996; Lary et al., 1996; Randeniya et al., 1997; Erle et al., 1998).

After large volcanic eruptions, the aerosol loading in the stratosphere and the surface area densities (hereafter SAD) available for reaction (1) to occur are dramatically enhanced (e.g. Deshler et al., 2003). As a result, the amount of ozone-depleting $NO_x$ is strongly reduced (e.g. Prather, 1992; Johnston et al., 1992; Fahey, 1993; Mills et al., 1993; Solomon et al., 1994; Kondo et al., 1997; Sen et al., 1998) whereas $HNO_3$ amounts increase (Koike et al., 1993; Webster et al., 1994; Koike et al., 1994; Rinsland et al.,



2003) as shown for the Pinatubo aerosols. Different impacts on stratospheric ozone are expected
depending on the altitude. In the middle stratosphere (above ~30 hPa) where ozone loss is dominated by
$NO_x$, the presence of volcanic aerosols can result in layers of increased net production of ozone due to
the suppression of the $NO_x$ cycle by the $N_2O_5$ hydrolysis (Hofmann et al., 1994; Bekki and Pyle, 1994;
Tie and Brasseur, 1995). In the lower stratosphere, halogen ($ClO_x$ and $BrO_x$) and hydrogen ($HO_x$)
radicals play a dominant role in ozone depletion and their abundances, which depend on $NO_x$ levels, are
increased (in particular for halogen species, as the rate of gas-phase conversion of ClO into the $ClONO_2$
reservoir is reduced), resulting in an enhanced catalyzed ozone loss (McElroy et al., 1992; Granier and
Brasseur, 1992; Brasseur and Granier, 1992; Hofmann et al., 1994; McGee et al., 1994; Bekki and Pyle,
1994; Salawitch et al., 1994a; 2005; Tie et al., 1994; Solomon et al., 1996; Solomon, 1999).
However, the $NO_x$-to-$HNO_3$ conversion by reaction (1) shows saturation as the aerosol SAD
increases because the amount of $N_2O_5$ present in the stratosphere is limited by its production rate by the
gaseous reaction $NO_2 + NO_3$ (Fahey, 1993; Prather, 1992; Mills et al., 1993; Tie et al., 1994; Solomon
et al., 1996; Kondo et al., 1997; Sen et al., 1998). Consequently, ozone loss rates are expected to be
limited because the saturation of the $NO_x/NO_y$ response to the aerosol increase dampens the increase in
$ClO/Cl_y$ (Fahey et al., 1993; Tie et al., 1994). Reaction (2) does not show such a rapid saturation resulting
in enhanced ozone depletion by chlorine catalytic cycles in cold air masses as the aerosol loading
increases (Fahey et al., 1993). The $BrONO_2$ hydrolysis through reaction (3) is primarily dependent on
the aerosol loading and is enhanced in periods of high volcanic aerosol loading. The resulting increase
of $BrO_x$ and $HO_x$ radical concentrations and decrease in HCl (due to enhanced OH) accompanied by an
increase in $ClO_x$ radicals is expected to give further ozone loss in the lower stratosphere at all latitudes
and seasons (Lary et al., 1996).
The year-to-year variability of ozone at northern mid-latitudes appears closely linked to changes in
chlorine partitioning driven by volcanic aerosols from major eruptions, with stronger effects than solar
cycle contributions on the mid-latitude ozone depletion (Solomon et al., 1999 and references therein).
This is expected in periods with a stratosphere perturbed by elevated chlorine levels from anthropogenic
activities (Tie and Brasseur, 1995; Solomon et al., 1996). In the past decade no event comparable to the
Pinatubo or El Chichon eruptions was observed. However, several volcanic eruptions, though of much
lesser amplitude, impacted the aerosol burden in the lower stratosphere over periods of months (Vernier
et al., 2011). These "moderate" eruptions have occurred in a period of still high chlorine loading with
potential impact on stratospheric ozone chemistry.
In this paper, we study the chemical impact of a short-term change in the amount of stratospheric
sulfate aerosols resulting from one of these "moderate" volcanic eruptions on some key aspects of
stratospheric chemistry and on ozone loss. The eruption of the Sarychev volcano on 15 and 16 June
2009 provides a very good opportunity to conduct such an investigation because 0.9 Tg of sulfur dioxide
were injected in the lower stratosphere (Clarisse et al., 2012) resulting in enhanced sulfate aerosol
loading and surface area densities up to 19 km for a period of about 8 months (Jégou et al., 2013). The
approach consists in analyzing the effect of the heterogeneous chemical reactions associated with
enhanced sulfate aerosol amounts on the lower stratospheric composition from balloon-borne
observations conducted in August-September 2009 from Kiruna in Northern Sweden (67.5°N, 21.0°E)
and aerosol-constrained simulations using a 3D Chemistry Transport Model (CTM). We show that at
the period of the measurements $N_2O_5$ has reformed and the role of its hydrolysis becomes important
again after the sunlit summer period. Here we estimate the ozone loss and discuss its amplitude in
comparison with the effect of the Pinatubo eruption.






## 2. Methodology

### 2.1 Balloon-borne observations

Our study is based on in situ and remote-sensing balloon-borne observations obtained during summer 2009 in Northern Sweden. More details about the instrument descriptions and retrieval techniques are given in the Appendix and in the references.

#### 2.1.1 In situ observations

Aerosol in situ measurements have been performed by the STAC (Stratospheric and Tropospheric Aerosol Counter) instrument which is an optical particle counter providing aerosol size distributions (Ovarlez and Ovarlez, 1995; Renard et al., 2008). This instrument has been used in a number of studies dedicated to the quantification of the aerosol content in the stratosphere at various locations and seasons (e.g. Renard et al., 2002; Renard et al., 2010). Eight vertical aerosol concentration profiles have been observed between August and September 2009 as reported by Jégou et al. (2013).

We focus here on the in situ vertical profiles of $N_2O$, $NO_2$ and $HNO_3$ provided by the SPIRALE (French acronym for SPectroscopie InfraRouge d'Absorption par Lasers Embarqués) infrared absorption spectrometer (Moreau et al., 2005) from two balloon flights. Firstly, the measurements during the 7 August 2009 flight (further on called SPIRALE-07082009) were conducted between 02:00 UT (04:00 local time) and 03:20 UT (05:20 local time) corresponding to altitudes of 14 km and 34 km respectively. The position of the balloon varied from 67.72°N-21.40°E to 67.63°N-20.92°E during the ascent. Secondly, the SPIRALE balloon flight on 24 August 2009 (furtheron SPIRALE-24082009, the measurements started at 21:00 UT (23:00 local time) at an altitude of 14 km and the maximum altitude of 34 km was reached at 22:30 UT (00:30 local time). The measurement position remained rather constant during the ascent with a displacement of the balloon from 67.91°N-21.09°E to 67.86°N-20.94°E. The data used in this study are averaged over a vertical range of 250 m (corresponding to ~1 minute of measurements).

#### 2.1.2 Remote-sensing observations

Since 1996 stratospheric $NO_2$ and $BrO$ have been measured by solar occultation by the DOAS balloon-borne instrument using the so-called Differential Optical Absorption Spectroscopy (DOAS) technique (e.g. Platt, 1994; Stutz and Platt, 1996; Ferlemann et al., 2000). The details of the vertical profile retrieval can be found in Butz et al. (2006) for $NO_2$ and in Harder et al. (1998), Aliwell et al. (2002), Dorf et al. (2006b) and Kreycy et al. (2013) for $BrO$. In our study we use the DOAS profile recorded in the stratosphere during the balloon ascent on 7 September 2009 between 15:15 UT (17:15 local time) and 16:35 UT (18:35 local time), corresponding to altitudes of 10 km and 30 km respectively.

The SALOMON (French acronym for "Spectroscopie d'Absorption Lunaire pour l'Observation des Minoritaires Ozone et $NO_x$") balloon-borne UV-visible spectrometer also uses the DOAS method to derive the mixing ratio profile of $NO_2$ (Renard et al., 2000; Berthet et al., 2002). SALOMON was initially based on the lunar occultation technique but on 25 August 2009, we flew a new version also able to use the Sun as direct light source to derive $BrO$ amounts. The profiles used in this study have been obtained on 25 August 2009 during solar occultation between 18:50 UT (20:50 local time) and 19:30 UT (21:30 local time). The float altitude was of 33 km and the position of the tangent point varied from 71.0°N-13.3°E to 71.4°N-12.6°E for altitudes below 19 km which are the main focus of our study as a result of the presence of the volcanic aerosols.

Variations of solar zenith angle (SZA) along solar occultation lines of sight and associated concentration variations are likely to impact the retrieved vertical profiles near sunrise and sunset especially below 20 km (Newchurch et al., 1996; Ferlemann et al., 1998). This effect can be corrected using a photochemical model (e.g. Payan et al., 1999; Harder et al., 2000; Butz et al., 2006). However, some retrievals from occultation measurements do not include corrections for diurnal variations in





concentrations because such corrections are strongly dependent on the photochemical model used in the
retrieval algorithm and are likely to result in additional errors (Randall et al., 2002).
In our study, the $NO_2$ profile from SALOMON instrument was recorded on 25 August 2009 from a
typical solar occultation at constant float altitude. Applying a photochemical correction to convert the
$NO_2$ concentrations to values expected at 90° SZA results in differences of only 3%. This calculation is
in agreement with the work of Payan et al. (1999) who have reported differences of less than 6% between
photo-chemically corrected and non-corrected profiles of $NO_2$. We note that Bracher et al. (2005) have
estimated larger diurnal variation effects, i.e. of about 10%. In the following the SALOMON
uncorrected profile is used for comparisons with model outputs. The vertical profile observed by the
DOAS instrument was recorded on 7 September 2009 with a different observation geometry, i.e. during
the balloon ascent. In this case applying a photochemical correction gives differences of 24% and the
model-measurement comparison must be done for an SZA = 90°. Photochemical effects on the BrO
profile obtained by the SALOMON instrument from solar occultation measurements are estimated to be
of 10% and are taken into account in the error estimation in accordance with the study of Ferlemann et
al. (1998). Photochemical changes in the BrO SCDs recorded during balloon ascent are small and the
DOAS BrO profile has not been corrected to 90°SZA (Ferlemann et al., 1998; Harder et al., 2000; Dorf
et al., 2006b).
**2.2 Model calculations**
The REPROBUS 3D CTM has been used in a number of studies of stratospheric chemistry involving
nitrogen and halogen compounds in particular through comparisons with space-borne and balloon-borne
observations (e.g. Krecl et al., 2006; Berthet et al., 2005; Brohede et al., 2007). It is designed to perform
annual simulations as well as detailed process studies. A description of the model is given in Lefèvre et
al. (1994) and Lefèvre et al. (1998), as well as in the Appendix.
In this study, REPROBUS was integrated from 1 October 2008 to 1 October 2009 with a horizontal
resolution of 2° latitude by 2° longitude. The ozone field was initialized on 1 April 2009 from the
ECMWF ozone analysis. We have conducted a REPROBUS simulation (hereafter called Ref-sim)
constrained with typical background aerosol levels inferred from the 2D model and used as reference,
namely without presence of volcanic aerosols. A simulation (hereafter called Sat-sim) has been set up
by prescribing time-dependent variations of the stratospheric sulfate aerosol content from 1-km vertical
resolution extinction measurements by the Optical Spectrograph and Infrared Imaging System (OSIRIS)
instrument onboard the Odin satellite. OSIRIS aerosol extinction data used in this study are the validated
version 5 retrieved at 750 nm (Bourassa et al., 2012). They compare well with the profiles inferred from
the STAC balloon-borne aerosol counter (Jégou et al., 2013) thus providing confidence in the use of the
data as a basis for consideration of time dependent changes of aerosol content. OSIRIS data have been
averaged daily and zonally over 10° latitude bins. A standard Mie scattering model (Van de Hulst, 1957;
Wiscombe, 1980; Steele and Turco, 1997) has been run to convert extinction values to $H_2SO_4$ mixing
ratios from parameters of log-normal unimodal size distributions provided by the STAC instrument and
used in the work of Jégou et al. (2013) in the Sarychev aerosol conditions. The derived 3D $H_2SO_4$ mixing
ratios have been then incorporated into the model over the period of presence of the Sarychev aerosols
in the northern hemisphere lower stratosphere, i.e. from the beginning of July 2009 onwards. The
simulation has been conducted until October 2009 because OSIRIS data at high latitudes are lacking
beyond this period due to decreasing solar illumination.
We have conducted another simulation (hereafter called Bal-sim) driven by aerosol observations with
a slightly different approach. The method here consists in adjusting the $H_2SO_4$ mixing ratios in the model
to reproduce the range of SADs observed by the STAC aerosol counter in summer (Jégou et al., 2013).
These observed SAD values are used as reference from the beginning of August until the end of the
model run and are homogeneously distributed for latitudes above 40°N. This simplification is supported
by the similar aerosol SAD values observed by Kravitz et al. (2011) at a different mid-latitude location
in November 2009, i.e. ~2 months after the STAC measurements as mentioned by Jégou et al. (2013).
Also, the $SO_2$ plume rapidly converts into aerosol sulfate, spreads out over the hemisphere and appears
rather uniformly distributed from about the end of July (Haywood et al., 2010). Our computed
uniformity of enhanced levels of SAD from August to September 2009 is representative, at least to some



extent, of the geographical distribution of the optical depth signal observed by the CALIOP/CALIPSO
space-borne lidar over the northern hemisphere for this period (O'Neill et al., 2012). Note that in Bal-
sim, $H_2SO_4$ mixing ratios in July are taken from the Sat-sim simulation.
The simulation presented hereafter accounts for the standard deviations of aerosol SADs observed
from the balloon-borne STAC instrument and shown in **Figure 1**.

## 3. Impact of the volcanic aerosols on stratospheric nitrogen compounds: comparisons between balloon-borne observations and model simulations

### 3.1 Impact of transport on simulated $N_2O$ and $NO_y$

It has been shown that wind fields from meteorological analysis produce an excessively strong
Brewer-Dobson circulation (BDC) in the stratosphere (e.g. Legras et al., 2005; Monge-Sanz et al., 2007)
which affects the ability of CTMs to represent the global distribution of long-lived tracers. Past model
calculations used to significantly underestimate $NO_x$ and $NO_y$ concentrations (e.g. Sen et al., 1998; Gao
et al., 1999; Wetzel et al., 2002; Stowasser et al., 2003) and Berthet et al., (2006) mainly attributed this
problem to transport calculation issues for $N_2O$. Following the work of Legras et al. (2005), REPROBUS
has been driven by 3-hourly ECMWF wind fields obtained by interleaving operational analysis and
forecasts. Using these more timely resolved and less noisy ECMWF wind fields reduced the ascent
velocities of the upward branch of the Brewer-Dobson circulation in the tropics, largely reduced the
model-measurement discrepancies by increasing the simulated global $NO_y$ and $NO_x$ amounts from
increased $N_2O$ photo-oxidation (Berthet et al., 2006). In this configuration, the summer 2009
REPROBUS simulations are in agreement with the SPIRALE in situ observations, especially at the
altitudes of the Sarychev aerosols (**Figure 2**).
The effect on simulated total $NO_y$ can be investigated by converting the vertical profile of $N_2O$,
following the strategy of Berthet et al. (2006) based on $N_2O$-$NO_y$ correlation curves. Since the study of
Michelsen et al. (1998) global emissions of $N_2O$ have increased and therefore the $N_2O$-$NO_y$ correlation
curve reported therein needs some revision. As a consequence, we have constructed updated high-
latitude $N_2O$-$NO_y$ correlation curves from the IMK/IAA V5R_220 MIPAS-Envisat data for the high-
latitude in summer stratosphere (Fischer et al., 2008; data available at http://www.imk-
asf.kit.edu/english/308.php) as shown in **Figure 3** in which the Michelsen et al.'s former results are also
represented for comparison. The estimated vertical profile of $NO_y$ (hereafter $NO_y$*) derived from the
conversion of the in-situ profile of $N_2O$ using the MIPAS correlation curve is presented in **Figure 2**.
Above 25 km, the $NO_y$* profile presents a non-monotonous trend in comparison with the $NO_y$ profile
computed by the 3D version of REPROBUS, since in the model the vertical structures on the observed
$N_2O$ profile are amplified by the conversion to $NO_y$* through the $N_2O$-$NO_y$ correlation. Above about 20
km $NO_y$* is almost systematically lower than the 3D REPROBUS $NO_y$ profile whereas better overall
agreement is observed for the volcanic aerosol loaded lower stratosphere.

### 3.2 Photochemical conditions

$N_2O_5$ is produced mainly at night from the recombination of $NO_2$ with $NO_3$ and destroyed during the
day by photolysis leading to the reformation of $NO_2$. $NO_3$ is formed mainly at night by the reaction of
$NO_2$ with $O_3$. The summer season provides particular conditions for stratospheric $NO_y$ chemistry. In this
period, some regions of the polar stratosphere receive continuous solar illumination for many weeks
which results in permanent photolysis reactions and enhances conversion of nitrogen reservoirs ($N_2O_5$
and $HNO_3$) to $NO_x$. Decreases of $HNO_3$, the major $NO_y$ species at mid- and high latitudes are manifest



in observations (Santee et al., 2004; Lindenmaier et al., 2011) and models (Chipperfield, 1999). With
the onset of continuous photolysis in high-latitude air masses, $N_2O_5$ production (occurring significantly
at night) stops abruptly because $NO_3$ amounts are kept low due to rapid photolysis, thereby preventing
$N_2O_5$ formation as shown on **Figure 4** above the Esrange/Kiruna balloon launching base. $N_2O_5$
hydrolysis ceases as well and the $NO_x/NO_y$ ratio becomes primarily controlled by gas-phase reactions,
$NO_x$ being principally destroyed by $NO_2 + OH$ reaction and produced by $HNO_3 + OH$ reaction and
photolysis of $HNO_3$ (Osterman et al., 1999; Dufour et al., 2005). A period of enhanced conversion of
$NO_y$ to $NO_x$ occurs until about beginning of August (Brühl et al., 1998) as reflected in **Figure 4**.
Consequently, $NO_x$ becomes the principal catalyst for ozone loss with local destruction rates which can
exceed 0.3% per day in summer air masses (Fahey and Ravishankara, 1999).
**Figure 4** shows the recovery of $N_2O_5$ on the return of sunset at high latitude (around day 213 at the
beginning of August for the considered location) around 17.5 km. When $NO_3$ reforms at the beginning
of August, significant conversion of $NO_2$ to $N_2O_5$ occurs during the night. The associated decrease in
$NO_x$ is reflected in **Figure 4**. The conversion of $N_2O_5$ to $HNO_3$ through reaction (1) occurs almost
exclusively at night. As the season progresses, the increase in the conversion rate caused by the increase
in night duration is moderated by the decrease in $NO_2$ amounts at the beginning of the night. As
expected, increasing SAD values in the model to reproduce the volcanic aerosol levels has no effect on
$N_2O_5$ (and on the production of $HNO_3$) and on $NO_x$ during the period of continuous solar illumination.
However, from the onset of $N_2O_5$ recovery a significant decrease in the $N_2O_5$ and $NO_x$ levels in
comparison with the background aerosol simulation is calculated as the lifetime of $N_2O_5$ in reaction (1)
is reduced (e.g. Kinnison et al., 1994) and as further nitrogen oxides are converted to the more stable
$HNO_3$ reservoir.
This situation implies that the balloon flights performed from August 7, 2009 match the
photochemical conditions for which volcanic aerosols likely had an impact on $NO_y$ partitioning via
elevated $N_2O_5$ hydrolysis. Some variability in modelled $N_2O_5$ (**Figure 4**) is due to the effect of
meridional transport to high latitudes which can be an important factor setting the stage of the chemical
conditions at the measurements location.
**3.3 $NO_2$**
**3.3.1 Model comparisons with observations**
In a stratosphere impacted by enhanced aerosol loadings after major volcanic eruptions, $NO_x$ amounts
are expected to be linked to aerosol concentrations. Observations of the $NO_2$ column has shown strong
anti-correlation with increasing aerosol amounts in mid-latitude conditions in spring (Mills et al., 1993).
In polar summer, strong reductions of $NO_x$ amounts have been observed in the presence of the Pinatubo
aerosols as a result of enhanced $N_2O_5$ hydrolysis (e.g. Solomon et al., 1994). For the Sarychev situation,
minima in $NO_2$ concentrations appear closely correlated with enhancements in aerosol amounts in the
lower stratosphere (**Figure 5**). Thus the empirical evidence supports the view that $NO_x$ chemistry is
largely driven by heterogeneous processes even in the case of a moderate volcanic eruption. Here
reductions in expected $NO_2$ of up to a factor of ~2 is seen for aerosol increases of ~3 (with respect to
the mean profiles). Conversely, layers with lower aerosol amounts, i.e. not affected by transport of the
volcanic aerosols, show maximums in $NO_2$ concentrations.
Model simulations have been conducted to provide further insight into the chemical impact of the
volcanic aerosols on $NO_x$ and $NO_y$ partitioning and to compare with several balloon-borne observations.
**Figure 6** presents the measured in-situ profiles of $NO_2$ obtained for two different cases of photochemical
conditions, i.e. for SPIRALE-07082009 around 02:15 UT, at ~87° SZA, and SPIRALE-242009, around
21:15 UT at a SZA of ~100°, together with REPROBUS model outputs for altitudes below 20 km where
the Sarychev aerosols were present. The reference simulations (i.e. without volcanic aerosols)
significantly overestimate the $NO_2$ observations with differences as large 56-57% (values with respect
to the measured profile) between 14 and 19 km for SPIRALE-07082009 and SPIRALE-24082009
(**Table 1**). The model results have also been assessed by the remote sensing observations from the
SALOMON and DOAS instruments flown on 25 August and 7 September 2009 respectively. Non-
volcanic model calculations show also discrepancies with solar occultation measurements in the lower





stratosphere (**Figure 7**), where the model overestimates measured $NO_2$ by 51% and 75% for the SALOMON flight on 25 August 2009 and the DOAS flight on 7 September 2009, respectively (**Table 1**).

The embedded plots in **Figure 6 and Figure 7** show the comparison above the Sarychev aerosol layer, i.e. for the whole range of altitudes observed by the instruments (up to ~35 km). Calculated $NO_2$ amounts overestimate the observations by 23% and 15% on average above 20 km for the SPIRALE-07082009 and SPIRALE-07082009 simulations, respectively. These values suggest that the model-measurement differences in the lower stratosphere may be only partly attributed to remaining uncertainties in calculations of transport. This issue is further investigated in section 3.6. Above 20 km the simulated profiles show good agreement with SALOMON and DOAS observations, except above 30 km for the flight on 25 August 2009.

The Sat-sim simulations driven by the aerosol content inferred from OSIRIS satellite data show significant improvement in comparison with the non-volcanic calculations, the model outputs matching well the $NO_2$ observations with model-measurement differences of 5-7% (in absolute values) for all dates (**Table 1**). Likewise, the measured $NO_2$ profiles and the model results obtained from the Bal-sim constrained by a range of aerosol SADs observed by the STAC aerosol counter show good agreement with for instance average differences of 3±20% and -16±20% for the SPIRALE-07082009 and SPIRALE-24082009, respectively. It can be noted that the REPROBUS calculations do not reproduce some of the vertical structures detected by the SPIRALE instrument, i.e. between 17.5 and 19.5 km for SPIRALE-07082009 and at 17 km and 20.5 km for SPIRALE-24082009. This is likely due to vertical resolution of the model or inaccurate simulation of mixing effects in the CTM as already mentioned in previous studies showing this kind of comparisons (e.g. Berthet et al., 2006). We note also that all simulation results deviate from the lower altitude points in the SALOMON and DOAS profiles. Part of this discrepancy might be due to effects of possible concentration inhomogeneities along the lines of sight which are likely to induce biases in the retrieved profiles from remote sensing instruments especially in the lower stratosphere (Berthet et al., 2007).

Calculated differences between the volcanic-aerosol-constrained and the reference simulations provide an estimation of the chemical perturbation induced by the Sarychev aerosols. Considering together the results from OSIRIS and balloon-driven simulations, reductions in $NO_2$ mixing ratios between 31 and 47 % are simulated on average below 19 km (**Table 2**). The similar $NO_2$ reduction for SPIRALE-07082009 and SPIRALE-24082009 once again indicates that enhanced hydrolysis of $N_2O_5$ onto volcanic aerosols is efficient even for conditions of incomplete recovery of $N_2O_5$. For a stratosphere affected by the Pinatubo aerosols, decreases ranging from 30 to 45% have been reported both in model calculations of $NO_2$ concentrations (Kinnison et al., 1994; Webster et al., 1994) and in the $NO_2$ columns (Johnston et al., 1992; Koike et al., 1993; Koike et al., 1994; Solomon et al., 1994). At a glance, the amplitude in the $NO_2$ reduction is therefore similar for both eruptions but it should be noted that results from these above-mentioned studies were provided for different latitudes, various seasons and correspond to wider altitude ranges as a result of the larger vertical extent of the Pinatubo aerosol cloud.

### 3.3.2 Saturation effect of $NO_x$ reduction

The reduction of $NO_x$ from the results described above is significant but also indicates some saturation through reaction 1 for the range of SAD observed for the Sarychev aerosols. The partitioning between $NO_x$ and $NO_y$ is expected to become insensitive to increases in aerosol SAD beyond a certain value when $N_2O_5$ hydrolysis is the dominant sink for $NO_x$ because the nighttime formation of $N_2O_5$ by reaction of $NO_2$ and $NO_3$ is quadratically dependent on $NO_x$. This effect is reflected in **Figure 8** presenting the $NO_2$ amounts versus aerosol SAD as observed by the SPIRALE instrument and simulated by the REPROBUS CTM. $NO_2$ reduction shows a kind of asymptotic behaviour as the heterogeneous rate of reaction 1 becomes large with increasing aerosol SAD. In this case 82-88% of $NO_y$ are stored in $HNO_3$. We must keep in mind that Figure 8 does not accurately demonstrate the saturation effect because our NOx-SAD curve has been constructed for a wide range of altitudes (i.e. with different NOy amounts and photochemistry) and not for a constant level. Observations obtained separately for different aerosol loadings but similar in terms of altitude levels and SZA would have been necessary to point out a clear asymptotic value, as a proof of the saturation effect. Nevertheless, our results indicate saturation for SAD values larger than about 4 $\mu m^2.cm^{-3}$ which is reached on average for altitudes around 18 km. The



net reduction of $NO_x$ reported for the Pinatubo aerosols tends to saturate at similar SAD values in the
18-22 km range, as shown in the works of Fahey et al. (1993), Kondo et al. (1997) and Sen et al. (1998).
**3.4 $HNO_3$**
**Figure 9** compares the $HNO_3$ profiles observed by the SPIRALE instrument and the simulations by
the REPROBUS CTM for SPIRALE-07082009 and SPIRALE-24082009. We show here model outputs
for total $HNO_3$ (i.e. both in the gas phase and condensed), but note that because $HNO_3$ is rapidly released
into the gas phase, gaseous $HNO_3$ would give the same results.
In the lower stratosphere, the simulated reference profiles for $HNO_3$ are mostly within the errors bars
of the measurements (calculated model-measurement differences are of -3% and -4% below 19 km for
SPIRALE-07082009 and SPIRALE-24082009, respectively as shown in **Table 1**) though some specific
vertical structures are not reproduced by the model. The agreement is even good up to 35 km confirming
that transport may not be a major issue in the comparisons.
Results from the volcanic-aerosol simulations appear also within the measurement error bars
(calculated model-measurement differences are between 6 and 8% for the Sat-sim results for SPIRALE-
07082009 and SPIRALE-24082009, respectively as shown in **Table 1**). The production of $HNO_3$ by
heterogeneous chemistry generally appears less effective in the lower stratosphere than above 20 km
when volcanic aerosols are present (see figure 3 of Webster, 1994; Plate 3 of Danilin et al., 1999).
However, the production of $HNO_3$ in the lower stratosphere is considered as significant for the Sarychev
derived aerosols because the simulations including volcanic aerosols increase simulated $HNO_3$ amounts
by 9-11% below 19 km as indicated by the Sat-sim results (**Table 2**). Simulated reduced levels of $NO_x$
correspond to the produced additional amounts of $HNO_3$. For instance, the simulated $NO_x$ decrease of -
0.21 ppbv matches the +0.22 pbbv increase of $HNO_3$ at 16 km for SPIRALE-07082009. Note that in
this context, the role of the $NO_2$ + OH reaction with respect to $NO_x$ conversion by enhanced $N_2O_5$
hydrolysis on the detailed partitioning between $NO_x$ and $HNO_3$ is not so clear (Coffey and Mankin,
28 1993).
After the Pinatubo eruption, maximum $HNO_3$ column increases of 30-40% were measured at mid-
latitudes (Koike et al., 1994). When Pinatubo aerosol SADs had decreased to values comparable to the
summer 2009 SADs around 16 km, i.e. 7.5-9 $\mu m^2.cm^{-3}$ in fall 1993 (Berthet et al., 2002), the percent
change in the $HNO_3$ column had dropped below 20% (Koike et al., 1994; Rinsland et al., 2003). Overall,
this reported effect is larger than in our observations indicating a more limited production of
stratospheric $HNO_3$ after the Sarychev eruption. However, quantifying the difference between both
eruptions in term of chemical effects remains difficult as mentioned for $NO_2$. In particular, the observed
signature of the Pinatubo-induced $HNO_3$ enhancement was not limited to the lower stratosphere and was
prevailing above the 420-465 K (~16-18 km) vertical range (Webster et al., 1994; Santee et al., 2004).
**3.5 $NO_2/HNO_3$ ratio**
The uncertainty in the simulated $NO_y$ is expected to be minimized by considering the ratios of
individual components of $NO_y$ to total $NO_y$ as shown by Wetzel et al. (2002) for summer mid-latitude
conditions. When no measurements of total $NO_y$ are available, the $NO_2/HNO_3$ ratio can be used as a
good approximation to reduce the uncertainty in the model estimate of $NO_y$ (e.g. Webster et al., 1994;
Berthet et al., 2006). This is especially useful for SPIRALE-07082009 and SPIRALE-24082009 for
which modelled $NO_2$ and $HNO_3$ amounts account for more than 92% of total $NO_y$.
**Figure 10** presents the $NO_2/HNO_3$ ratios observed by SPIRALE in comparison with the REPROBUS
model simulations. The $NO_2/HNO_3$ ratio in the lower stratosphere is typically 0.2 but the measurements
indicate a smaller ratio. Under background aerosol loadings the observed low $NO_2/HNO_3$ ratios are not
matched by the reference simulation with the differences below 19 km being 62-63% for both flights
(**Table 1**). A good agreement is obtained between both measurements and the model by including the
Sarychev aerosols with absolute differences decreasing to 3±20% and 1% for the Balloon-aero-sim and
Satellite-aero-sim simulations, respectively, for SPIRALE-07082009. No clear improvement can be
noticed from the model-measurement comparisons of the $NO_2/HNO_3$ ratio and the both species (**Figures**



**4** and **7**). Again this indicate that uncertainties in transport calculation are not the main explanation for
the model-measurement discrepancy observed for the lower stratosphere.
The Sat-sim aerosol constrained simulations of the reduction in the $NO_2/HNO_3$ ratio are 36% and
44% for SPIRALE-07082009 and SPIRALE-24082009, respectively (**Table 2**). These ratios are similar
to the Bal-sim outputs. For the Pinatubo aerosol loaded stratosphere, comparable reductions ranging
from 20 to 45% have been reported both in the observed $NO_2/HNO_3$ column ratios (Koike et al., 1994)
and in model calculations (Webster et al. (1994).
**3.6 One-dimensional model calculations**
The measurement-model comparisons still show some discrepancies above 20 km. A way to discard
a possible remaining effect of transport and further improve the modelling of total $NO_y$ is to use
constrained one-dimensional (1D) calculations (Dufour et al., 2005; Berthet et al., 2006). These may
allow us to refine the quantification of the enhanced heterogeneous processes resulting from the
Sarychev eruption. Following the strategy of Berthet et al. (2006), the 1D-REPROBUS initialization is
constrained by available $N_2O$ observations (here the profile measured by SPIRALE) and by the
corresponding $NO_y*$ profile as illustrated in **Figure 2**.
The 1D reference simulation (Ref-sim 1D) is computed with background aerosol levels, while the
Sarychev aerosol affected simulation is constrained with the mean aerosol profile measured during the
balloon-borne observations (Bal-sim 1D). As a result of the $NO_y*$ input in the calculations, the Ref-sim
1D simulations show very good agreement for $NO_2$ and $HNO_3$ (and consequently for $NO_2/HNO_3$) with
the SPIRALE measurements above 20 km, thus mostly leading to better results than the 3D reference
simulation (**Figures 6, 9 and 10**) especially for $NO_2$ and $NO_2/HNO_3$. At the altitudes affected by the
Sarychev aerosols, the model results deviate from the observed profile. Here differences are 48-60%
(absolute values) for the $NO_2/HNO_3$ ratio for SPIRALE-07082009 and SPIRALE-24082009 (**Table 1**).
We note that fine structures in the measured profile are not reproduced by the 1D model as a matter of
height resolution and interpolation (Berthet et al., 2006). As for the 3D model profiles, the 1D
simulations constrained by observed aerosol quantities (Bal-sim 1D) are in good agreement with the in
situ measurements with the calculated model-observation differences being 12-16% (absolute values)
for the $NO_2/HNO_3$ ratio (**Table 1**). Overall the 1D simulations including volcanic aerosol loadings do
not show evidence of significant improvement in the comparisons. These results confirm that the model-
observations differences in the lower stratosphere can be mostly attributed to heterogeneous processes
and not to spurious calculations of transport.
The calculated chemical impact of the Sarychev aerosols on $NO_2$ and $HNO_3$ gives percentage values
comparable to the 3D simulation results, with for instance a reduction by 22 and 34% in $NO_2$ amounts
for SPIRALE-07082009 and SPIRALE-24082009, respectively (**Table 2**).
**4. Impact of the volcanic aerosols on the coupled catalytic cycles involving**
**halogen, nitrogen and $HO_x$ compounds**
**4.1 Chlorine partitioning**
Several studies have revealed the impact of the Pinatubo eruption on the stratospheric halogen
chemistry. This has been shown to be of particular importance regarding ozone destruction processes
through the partitioning of chlorine reservoir species and activation of chlorine radicals on volcanic
aerosols (e.g. Solomon, 1999 and references therein). Some volcanic eruptions are likely to inject
halogenated compounds within the stratosphere therefore impacting directly the halogen content and


bypassing (or adding to) in situ heterogeneous processes. For the Sarychev volcano eruption, an injection
of several ppbv of HCl in the stratosphere has been reported by Carn et al. (2016) using Microwave
Limb Sounder (MLS) data, mainly below the 140 hPa level (see their Figure 4). However the exact
altitude of injection is inaccurate because of the low vertical resolution of MLS data, i.e. ~3 km. In
addition, MLS HCl measurements are known to be biased high below the 100 hPa level and are not
recommended for scientific use (Livesey et al., 2011) making difficult to infer a robust injection amount.
HCl amounts in the lower stratosphere has returned to background levels within about two weeks (Carn
et al., 2016). No difference with respect to background HCl levels is apparent above Kiruna and over
the Northern hemisphere in July and at the period of the balloon campaign, indicating fast dilution of
the HCl plume after the eruption (MLS data available at http://giovanni.gsfc.nasa.gov/giovanni/ and
http://disc.sci.gsfc.nasa.gov/). Thus no effect is expected on the total inorganic chlorine in our model
calculations.
We therefore examine the direct impact of the Sarychev sulfate aerosols on the chlorine partitioning
in connection with $NO_x$ and $HO_x$ in the lower stratosphere. Heterogeneous reactions on volcanic aerosols
involving the $ClONO_2$ and HCl chlorine reservoirs (especially reaction 2) have been shown to play a
major role in determining the abundance of active chlorine and therefore they are likely to compete with
reaction 1 as a sink of $NO_x$ depending on ambient temperature values (e.g. Hanson et al., 1994).
Significant decreases of HCl and corresponding increases in $ClONO_2$ have been reported for
temperatures below 210 K in the lower stratosphere with a strong temperature sensitivity when volcanic
aerosol amounts are large (Michelsen et al., 1999; Webster et al., 1998; Webster et al., 2000). **Table 3**
presents the calculated effects of the Sarychev aerosols on the partitioning of the halogen species at 16.5
km. Simulated levels of HCl decrease by 3% (~20 pptv) which is much smaller than the change observed
by Webster et al. (2000) for the Pinatubo aerosols (about -31% at 21 km). Higher levels of $ClONO_2$ are
simulated post the Sarychev eruption with respect to background conditions with increases of about 16%
(~20 pptv). ClO and HOCl increase by 106% (~6 pptv) and 217% (~2 pptv) respectively at daytime. It
is interesting to notice that these results for ClO are comparable to the calculations of Tie et al. (1994)
who show ClO increases by at least 5 pptv in the lower stratosphere for summer 1992 at a time when
Pinatubo related aerosol SADs were similar to August 2009 values.
The impact of the volcanic aerosols on the chlorine partitioning appears somewhat small since it is
primarily the consequence of the increasing losses of HCl by enhanced OH through reaction HCl + OH
→ Cl + $H_2O$ (McElroy et al., 1992; Webster et al., 2000) rather than by reaction 2 for which the
efficiency is low in the ~215-225 K range of temperatures mostly encountered in the lower stratosphere
over the August-September 2009 period (see Figure 9 in Jégou et al., 2013). In fact, in the model $HO_x$
is increased by 51% (~1.4 pptv) (**Table 3**) and destruction of HCl by OH is faster than the HCl formation
reaction Cl + $CH_4$ → HCl + $CH_3$. An additional source of OH may be due to photolysis of $HNO_3$
(Rodriguez et al., 1991; Webster et al., 2000). Also the decreased reaction rate of reaction $NO_2$ + OH +
M → $HNO_3$ + M in reduced $NO_x$ conditions (Kinnison et al., 1994) may increase OH. As also described
by Bekki and Pyle (1994), subsequent production of reactive chlorine and increase in ClO is
accompanied by an increase in $ClONO_2$ amounts through increased rate of reaction ClO + $NO_2$ + M →
$ClONO_2$ + M, for which ClO is the limiting reactant. To a lesser extent, decreased rate of reaction 3 for
the observed temperature range contributes to this increase. Overall, the $ClONO_2$ increase compensates
for the HCl decrease in reaction 3 (Kinnison et al., 1994; Michelsen et al., 1999; Webster et al., 2000).
HOCl amounts rise as a result of slightly enhanced $ClONO_2$ hydrolysis and production by enhanced
$HO_x$ through reaction $HO_2$ + ClO → HOCl + $O_2$.
**4.2 Bromine compounds**
**4.2.1 Effect on BrO**
Coupling between chlorine and bromine compounds is of particular importance in the lower
stratosphere and the role of bromine chemistry in regulating chlorine partitioning must be considered
(e.g. Lary et al., 1996; Erle et al., 1998; Salawitch et al., 2005). Heterogeneous bromine reactions are



expected to increase the coupled gas phase ClO/BrO catalytic ozone destruction cycles. Because
$BrONO_2$ hydrolysis (reaction 4) is not temperature dependent, its effects on the chemistry of the lower
stratosphere are primarily dependent on the aerosol loading and not on latitude or SZA (Lary et al.,
1996; Kondo et al., 1997; Erle et al., 1998).
Some incidents of a direct injection of bromine into the stratosphere by volcanic eruptions have been
reported. The study of Hormann et al. (2013) based on space-borne observations of BrO however
indicate that stratospheric injection of bromine was insignificant after the Sarychev eruption. We
therefore expect that stratospheric bromine chemistry was only modified by the enhanced aerosol
loading (e.g., Lary et al., 1996). BrO was the only key halogenated radical detected during the summer
2009 balloon campaign. Vertical profiles were provided by the SALOMON and DOAS instruments on
25 August 2009 and 7 September 2009 respectively (**Figure 11**). They were simultaneously measured
with the $NO_2$ profiles presented in section 3.2.2. When volcanic aerosol SAD are included BrO amounts
are increased in the lower stratosphere, matching the observations within the error bars (**Figure 11**).
Differences between the model and the observations and between the various simulations are
summarized in **Table 1** and **Table 2** respectively.
Simulated results related to the bromine chemistry at 16.5 km are presented in **Table 3** for the
August-September 2009 period. At daytime part of the BrO enhancement is linked to the decreased loss
by the three body reaction with decreased $NO_2$. The other part is expected to be controlled by $BrONO_2$
hydrolysis which is by far the most efficient bromine heterogeneous reaction in the temperature range
observed in our study (Hanson and Ravishankara, 1995; 1996). Under high aerosol loading the rate of
the $BrONO_2$ hydrolysis is likely to compete with the $BrONO_2$ photolysis and with other gas phase
reactions which normally control the bromine partitioning at daytime (Lary et al., 1996). Here note that
the conclusion of Kreycy et al., (2013) on a possibly larger ratio of the photolysis and the three body
formation reaction for $BrONO_2$ (J(k)) than compiled by Sander et al., (2011) is not by affected by the
presence of the Sarychev aerosols in the lower stratosphere, since they addressed solar occultation
observations for solar zenith angles < 92.5° at 31 km (i.e., tangent heights > 24 km). After sunset
$BrONO_2$ production is ceasing and its enhanced hydrolysis on volcanic aerosols leads to strongly
increased formation of HOBr (+3.9 pptv or +141%) at an early stage of the night so that little $BrONO_2$
remains before dawn. This conversion at nighttime results in further release of OH and Br atoms in the
morning through photolysis of HOBr.
However, it is not clear if $BrONO_2$ hydrolysis is mainly responsible for the increase in BrO within
the lowermost stratosphere. Dedicated simulations to estimate the respective contribution of gas-phase
chemistry and heterogeneous processes on the control of BrO production under volcanic conditions have
thus been performed. The effects of the Sarychev aerosols on each chemical compound are calculated
by switching off reaction 4 and compared in terms of percentage differences with the simulations
including all chemistry. Results are summarized in **Table 3**. It particularly shows that under the
Sarychev aerosol loading, 18% of the daytime BrO production (+0.9 pptv or +22% at 16.5 km during
the August-September 2009 period when volcanic aerosols were present) is due to $BrONO_2$ hydrolysis.
This results implies that bromine chemistry in the gas phase coupled to processes controlling the $NO_y$
partitioning mainly govern BrO amounts (e.g., Lary et al., 1996).
**4.2.2 Role of $BrONO_2$ hydrolysis on other compounds**
As shown in **Table 3** for an altitude of 16.5 km, at night $BrONO_2$ amounts are mainly affected by
reaction 4 which controls 98% of its decrease under volcanic aerosol influence. Nearly 100% of the
night-time HOBr production is due to $BrONO_2$ hydrolysis which accounts for 44% of the increase in
OH radical amounts from the subsequent photolysis of HOBr at dawn. Therefore, under volcanic
conditions enhanced $BrONO_2$ hydrolysis nearly matches the contribution of nitrogen chemistry (see
section 4.1) as a source of OH (e.g., Hanisco et al., 2001).
This additional release of OH radicals has significant consequences in the chemistry of the lower
stratosphere. In our study the reduction in $NO_x$ from $BrONO_2$ hydrolysis are small (less than 2%) as
well as the overall effects on nitrogen partitioning confirming the conclusions of Lary et al. (1996) and
Kondo et al. (1997). In contrast, there is substantial repartitioning of the active chlorine families. The
catalytic increase in OH due to the hydrolysis of $BrONO_2$ leads to a reduction in the HCl lifetime which
is primarily dependent on the aerosol loading (Tie and Brasseur, 1996). The additionally produced OH



converts further HCl to ClO and, ultimately, to $ClONO_2$. As shown in **Table 3**, ~60% of the HCl
decrease, 39% of the ClO increase and 66% of the $ClONO_2$ increase are due to reaction 4 under the
Sarychev aerosol loading, thus illustrating a significant enhancement of the coupling between the
stratospheric chlorine and bromine photo-chemistry.
**5. Stratospheric ozone**
**5.1 Chemical ozone change**
Several studies have demonstrated that the effect of the Pinatubo aerosols on stratospheric ozone
depletion at mid-latitudes is particularly significant in winter and spring. For instance, maximum ozone
losses of 20-30% were reported for the 12 and 22 km altitude range monitored at some mid-latitude
locations during 1993 winter and spring (Hofmann et al., 1994) whereas $O_3$ decreases of 10-15%
occurred for the total ozone column (McGee et al., 1994; Randel et al., 1995). For the mid-latitude total
ozone column Tie and Brasseur (1995) calculated reductions of the order of 6% in late winter/early
spring. Similar decreases of total ozone were simulated for the summer northern hemisphere by Brasseur
and Granier (1992).
It is interesting to estimate the stratospheric ozone depletion induced by the Sarychev eruption which
differs from the Pinatubo eruption in terms of aerosol loading, season and latitude of injection, and
aerosol residence time. As said above, the model does not directly calculate possible effects of aerosols
on stratospheric temperature and circulation. All our simulations use the same transport calculations,
whereas ozone loss from Pinatubo in the northern mid-latitudes can be both attributed to chemical and
transport (such as increased tropical upwelling) effects (e.g. Telford et al., 2009). In the following, we
therefore solely calculate the change in ozone due to photochemistry.
We then compare model simulations with enhanced and background aerosol levels (**Figure 12**).
Results indicate chemical reductions in ozone of a few percent following the eruption when aerosol
levels are computed from the OSIRIS space-borne data. Accumulated ozone depletion reaches its
maximum above Kiruna around mid-September with changes of -1.5% (-20 ppbv) and -2.5% (-25 ppbv)
at 16.5 km and 14 km, respectively. Similar ozone changes are simulated when the model is driven by
the lower values of aerosol loading taken from STAC in-situ observations whereas when maximum
aerosol values from the STAC instrument are used ozone depletion is -2.8% (-25 ppbv) and -4% (-35
ppbv) at 16.5 km and 14 km, respectively (not shown). We clearly see that the reduction increases with
decreasing altitude. Ozone depletion values close to the tropopause appear larger than in the lower
stratosphere. This conclusion must be taken cautiously because the model does not include detailed
influence of various other chemicals (especially organic compounds) entrained from the troposphere
into lower stratosphere.
**5.2 Chemical mechanisms for the ozone change in the lower stratosphere**
In the lower stratosphere ozone removal rates are mainly controlled by the $HO_x$ and halogen catalytic
cycles which have been found to typically account for 30-50% and 30% of the total ozone loss
respectively, in non-volcanic conditions (Portmann et al., 1999; Salawitch et al., 2005). The $NO_x$ cycles
play a relatively minor role in the direct removal of ozone in the lower stratosphere but, as a result of
the coupling among the $NO_x$, $HO_x$ and halogen cycles, the rate of ozone removal is still very sensitive
to the concentration of $NO_x$ (Wennberg et al., 1994; Gao et al., 1999; Portmann et al., 1999; Salawitch
et al., 2005). Through the reaction of $HO_2$ with NO ($HO_2 + NO \rightarrow NO_2 + OH$), the decreased $NO_x$
concentrations after the Sarychev eruption result in a larger $HO_2/OH$ ratio (as shown in **Table 3**) than
for background conditions ($HO_2/OH$ ratios typically ranging from 4 to 7). Because the photochemical





removal of ozone in the lower stratosphere is dominated by processes involving $HO_2$, catalytic ozone destruction by $HO_x$ cycles is likely to be amplified after volcanic eruptions (Wennberg et al., 1994; 1995) though ozone loss rates are limited due to the saturation of the $NO_x/NO_y$ response. After the eruption of Sarychev the effectiveness of halogen cycles is enhanced due to increased $ClO_x$ resulting from OH increase (**Table 3**) (as explained in section 4.1). As said above, heterogeneous reactions activating chlorine are strongly and non-linearly dependent on temperature, implying slow rates at the average mid-latitude temperature conditions (minimum values of 215 K) (Hanson et al., 1994; Webster et al., 1998; Michelsen et al., 1999). Under these conditions the simulated depletion in ozone is restrained similarly to the finding of Tie et al. (1994). Note that in their study ozone reduction was about 5% in the lower summer stratosphere when Pinatubo aerosol SADs were comparable to our observations.

Part of the ozone depletion can be related to the coupled $BrO_x/ClO_x$ cycle which is expected to be responsible for 20-25% of the halogen-controlled loss under non-volcanic aerosol conditions (Portmann et al., 1999; Salawitch et al., 2005). **Table 3** shows that the hydrolysis of $BrONO_2$ accounts for more than 22% of the ozone loss at 16.5 km after the Sarychev eruption. Reaction 4 acts as a source of OH and accordingly reduces the HCl lifetime. This reduction in HCl lifetime is accompanied by an increase in the $ClO_x$ concentration and thereby indirectly couples the atmospheric chemistry of chlorine and bromine to amplify the chlorine-mediated ozone depletion. Because the sticking coefficient for hydrolysis of $BrONO_2$ on sulfate aerosols is not temperature dependent, this effect occurs at all latitudes and seasons in the lower stratosphere during high aerosol loading periods (Lary et al., 1996; Tie and Brasseur, 1996).

# 6. Summary and conclusions

Our study provides key observations of the chemical perturbation in the lower stratosphere by the moderate Sarychev volcano eruption in June 2009. 3D and 1D CTM simulations are performed to interpret balloon-borne observations of some key chemical species made in the summer high-latitude lower stratosphere. The modelled chemical response to the volcanic aerosols is treated by comparing simulations using background aerosol levels and simulations driven by volcanic aerosol amounts inferred from balloon-borne and space-borne observations.

Quantifying the impact of volcanic aerosols on stratospheric ozone chemistry is difficult as chemical and dynamical (radiative) effects simultaneously occur (Pitari and Rizi, 1993; Robock, 2000). The model is a CTM driven by ECMWF off-line meteorological data and does not describe radiative processes. In other words, volcanic aerosol radiative effects are not directly interactive with the circulation computed by the model. Radiative processes from the injection of volcanic aerosols in the tropics have been shown to have an impact on mean meridional circulation and ozone transport (Brasseur and Granier, 1992; Pitari et Rizi, 1993). In our study, effects of the Sarychev aerosols on mid-latitude stratospheric dynamics, if any, are at least at the first order intrinsically taken into account in the ECMWF analyses used for all simulations. REPROBUS does not take into account the aerosol impact on calculated photolysis rates which is likely to result in some differences between models when this process is computed or ignored (Pitari et Rizi, 1993; Pitari et al., 2014). However because the Sarychev eruption has impacted only mid-latitude lower stratosphere the effect on the photolysis frequency of molecular oxygen and ozone due to absorption and backscattering of solar radiation by the volcanic aerosols is expected to be very small in this region (Tie et al., 1994). Therefore, since all our simulations have been driven with the same wind and temperature fields our approach only estimates the chemical effects of the Sarychev aerosols.

The $NO_y$ chemistry appears to be very sensitive to the increase in SAD within the lower stratosphere resulting from the Sarychev eruption. A decrease in the $NO_x$ abundances is evident but shows some saturation as emphasized in a number of studies referring to cases of high sulfate aerosol loadings (e.g. Fahey et al., 1993). The effect of volcanic aerosols on nitrogen partitioning is also reflected in the





calculated production of $HNO_3$ as a result of the decrease of the $N_2O_5$ nitrogen reservoir from its
enhanced hydrolysis and $NO_x$ reduction.
Although direct comparisons in terms of solar illumination, latitude, injection altitudes and
temperature are not possible for distinct volcanic eruptions such as Pinatubo and Sarychev, it is
interesting to compare the effect of both eruptions on the photochemistry of the lower stratosphere.
Overall, although different in magnitude, the eruptions of Pinatubo and Sarychev show similar observed
and simulated depletion of $NO_2$, probably due to the saturation effect of the enhanced $N_2O_5$ hydrolysis.
In comparison with the Pinatubo period, the Sarychev aerosols led to less overall $HNO_3$ production in
the stratosphere possibly because the related $HNO_3$ enhancement has been shown to be considerably
weaker in the lowermost stratosphere (below ~18 km) than for sulfur injection into higher altitudes
(Webster et al., 1994; Santee et al., 2004). However, one must notice that previously reported modelling
studies on the Pinatubo aerosols were conducted with former chemical kinetic rate constants and
photolysis rates which have been largely updated ever since, somewhat adding complexity for
comparisons discussed within the present study.
For the Pinatubo aerosols, ozone destruction was not observed throughout the volcanic aerosol layer
because $N_2O_5$ hydrolysis reduced $NO_x$ related ozone loss, which even resulted in small increases of
ozone in the middle stratosphere (Bekki and Pyle, 1994; Tie and Brasseur, 1995). For the Sarychev
eruption, the volcanic aerosol layer is restrained to altitude levels below 19 km where the ozone
destruction processes by $HO_x$ and halogen catalytic cycles are expected to play a major role (e.g.
Salawitch et al., 2005) with some sensitivity towards $NO_x$ levels. To summarize, the increased
production of $HNO_3$ via $N_2O_5$ hydrolysis enhances the photolytic production of OH from $HNO_3$. As a
result, the gas-phase sink for HCl by reaction with OH is slightly enhanced and is associated with an
increase of ClO amounts. An important result from the heterogeneous hydrolysis of $BrONO_2$ is the
formation and subsequent photolysis of additional HOBr. The OH so produced additionally converts
HCl to ClO (and ultimately to $ClONO_2$). Accordingly, there is substantial repartitioning of the active
chlorine but effects of the $BrONO_2$ hydrolysis on nitrogen partitioning are insignificant. In this chemical
context, the magnitude of the ozone response to the Sarychev volcanic perturbation appears limited (i.e.
between -2.5 and -4% at 14 km considering the whole range of observed SADs) because the saturation
of the $NO_x/NO_y$ response limits the increase in $HO_x$ and in active chlorine (ClO) by enhanced $HO_x$,
precluding important ozone loss rates. Moreover, stratospheric temperatures remained too high (i.e.
mainly above 215 K) for efficient heterogeneous conversion of $ClONO_2$ to active chlorine, which could
have led to significant ozone depletion. For these temperature conditions, reaction 2 is not expected to
compete with $N_2O_5$ hydrolysis in the $NO_y$ partitioning (Fahey et al., 1993; Cox et al., 1994). Eventually,
the largest ozone destruction is restricted to the lowermost stratosphere (the bottom of the volcanic
aerosol layer close to the tropopause) where catalytic cycles are primarily controlled by $HO_x$ and where
the $NO_x$ photochemistry plays a very minor role.
However limitations in our model simulations also contribute to some model-measurement
discrepancies. A first major difficulty is to drive the model simulations with representative and
consistent inputs in term of volcanic aerosol loading. To address this issue, two different model runs for
aerosol forcing have been performed, one using OSIRIS satellite data converted to aerosol SAD fields
and the other one from in-situ balloon-borne observations. The OSIRIS satellite data represent zonally
and daily averaged values of SAD which may vary from a 3D construction based on the local surface
areas. The possible presence of aerosol streamers (geographical variations of the aerosol content)
resulting from the transport of the volcanic aerosols over the northern hemisphere present from mid-
July to September 2009 is likely to affect locally and regionally the $N_2O_5$ abundances and, to a lesser
extent, $NO_2$ and $HNO_3$ (Küll et al., 2002; Jucks et al. 1999). If our aerosol SAD dataset had been obtained
when the local concentrations were higher than the zonal mean values, then the calculated rate of the
heterogeneous reactions would be biased low and calculated $NO_x$ and $HNO_3$ abundances would be
systematically biased high and low respectively. This is not however evident in all our comparisons
from simulations based on OSIRIS aerosols. We also note that former studies mostly used 2D
simulations to investigate the chemical effects of the enhanced aerosol burden following the Pinatubo
eruption with some limitations in terms of meridional transport simulations. The second type of aerosol-
constrained simulation uses SADs from balloon-borne observed profiles. By definition, such in situ
observations deal with a particular location. Extrapolating in situ derived SADs to drive a 3D model at
a large scale may induce inaccurate simulations of the chemical impact of the aerosols (Kondo et al.,





2000). To account for this SAD-related uncertainty, our simulations based on in-situ data encompass the range of SADs derived from the STAC balloon-borne observations over the August-September 2009 period. Both satellite- and balloon-driven simulations give similar results in terms of $NO_2$ and $HNO_3$ amounts possibly because the in-situ observations represent well the aerosol loading at the northern mid-latitudes. Another explanation is that the saturation effect (roughly when SADs become larger than 3 $\mu m^2.cm^{-3}$) of the $NO_x/NO_y$ ratio is more relevant for the range of observed SADs than spatiotemporal inhomogeneities.

Secondly, adequate modelling of transport is also crucial for the partitioning of $NO_y$. Processes that control the vertical profiles of $NO_2$ and $HNO_3$ in the stratosphere are based on a complex interplay between dynamics and chemistry with the key issue to accurately simulate total $NO_y$ which may be not systematically achieved with 3D CTM calculations. Improved simulations of transport can be obtained by combing operational analyses with forecasts to construct 3-hourly meteorological data to drive the CTM (Berthet et al., 2006). We have applied this strategy in the present study. Using 1D modelling driven by in-situ observations or calculating $NO_2/HNO_3$ ratios to reduce transport effects does not clearly improve the model-measurement comparisons for the lower stratosphere. Although some features in the vertical profiles are not systematically captured by the model, this tends to indicate that the error in calculated transport is not large enough to account for the overall difference between measured and modelled $NO_2$ and $HNO_3$ when no volcanic aerosol loading is included in the model. Rather, these results show some evidence of the role of heterogeneous reactions at the surface of volcanic aerosols.

Thirdly, part of the discrepancies between model and observations might be attributed to spatial resolution issues. It may be tricky to compare model calculations with high resolution in-situ profiles and with remote sensing observations integrating over tens of kilometers (Berthet et al., 2007). For instance, discrepancies between remote sensing observations and model calculations have been reported for stratospheric $NO_3$ in case of localized temperature inhomogeneities as a result of the strong dependence of $NO_3$ cross sections and kinetics on temperature (Renard et al., 2001). $N_2O_5$ and $NO_2$ may be subsequently impacted because $NO_3$, together with $NO_2$, plays a central role in the equilibrium reaction controlling $N_2O_5$ in the gas phase.

In our study, no comprehensive sulfur chemistry is included in the model. We have also excluded dynamical and radiative effects on the ozone response which have been shown to be of primary importance when dense volcanic clouds are present (e.g. Pitari and Rizi, 1993; Kinnison et al., 1994; Tie et al., 1994). In a forthcoming study it would be interesting to compare dynamical/radiative and chemical effects of moderate volcanic eruptions on stratospheric ozone using Chemistry-Climate models with full sulfur chemistry and aerosol-dynamics interactive calculations.

Finally, it might interesting to investigate the effects of other volcanic plumes coming from moderate volcanic eruptions which are then transported to high-latitude regions when stratospheric temperatures are more favourable for chlorine activation and enhanced ozone loss (e.g. in winter). Activation of chlorine from volcanic sulfate aerosols and associated ozone depletion is arguably more significant in the cold temperature conditions of winter/spring, even above the formation threshold of Polar Stratospheric Clouds (Hanson et al., 1994). The eruption of the Calbuco volcano in the southern hemisphere in April 2015 could be a good candidate for the study of this process.



**Acknowledgements.**
The authors are grateful to the CNES (French acronym for "Centre National d'Etudes Spatiales")
balloon launching team for successful operations and the Swedish Space Corporation at Esrange. The
STRAPOLETE project and the associated balloon campaign has been funded by the French "Agence
Nationale de la Recherche" (ANR-BLAN08-1-31627), CNES, and the "Institut Polaire Paul-Emile
Victor" (IPEV). The study is supported by the French Labex « Étude des géofluides et des VOLatils–
Terre, Atmosphère et Interfaces - Ressources et Environnement (VOLTAIRE) (ANR-10-LABX-100-
01) managed by the University of Orleans. The ETHER database (CNES-INSU/CNRS) is partner of the
project. Further support for the DOAS balloon measurements came through the Deutsche
Forschungsgemeinschaft, DFG (grants PF-384/5-1 and 384/5-1 and PF384/9-1/2), and the European
projects EU projects Reconcile (FP7-ENV-2008-1-226365) and SHIVA (FP7-ENV-2007-1-226224).
We thank Michel Van Roozendael and Caroline Fayt from BIRA/IASB in Belgium for making available
the WINDOAS algorithm very well-adapted for data reduction methods based on the Differential
Optical Absorption Technique. We acknowledge the MIPAS/Envisat team from Karlsruhe Institute of
Technology (KIT) for making IMK/IAA data available.





**Appendix A: Technical description**
**A.1 The STAC aerosol counter**
Aerosol size distributions are provided in the 0.4–5 μm diameter size range (Ovarlez and Ovarlez,
1995; Renard et al., 2008). Since 2008, the number of available size classes has been increased from 7
to 14 within this size range (Renard et al., 2010). The counting uncertainty is obtained from the statistical
probability given by Poisson counting statistics (Willeke and Liu, 1976). This uncertainty, defined as
the relative standard deviation, is 60% for aerosol concentrations of $10^{-3}$ cm$^{-3}$, 20% for $10^{-2}$ cm$^{-3}$, and
6% for concentrations higher than $10^{-1}$ cm$^{-3}$. Laboratory comparisons between two copies of the STAC
aerosol counter using identical aerosol samples have shown differences of ±10% for concentrations
higher than $10^{-2}$ cm$^{-3}$. From these results, we define a measurement precision limited to ±10%. It should
be noted that comparisons with the aerosol concentrations measured by the University of Wyoming
optical particle counter (Deshler et al., 2003) have shown consistent results between both instruments
(Renard et al., 2002). STAC is calibrated in order to provide size distributions of non-absorbing liquid
aerosols which have been unambiguously observed in the 8–19 km altitude range in the case of the
Sarychev eruption (Jégou et al., 2013). Aerosol distribution moments are derived using well-known
analytical expressions. Using a statistical approach as described in Deshler et al. (2003), STAC counting
uncertainties (Poisson statistics and the ±10% precision) translate into uncertainties on distribution
moments, with estimated values of 40% for SAD. Profiles are typically averaged over a vertical range
of 250 m (corresponding to ~1 minute of measurements).
**A.2 The SPIRALE in-situ infrared spectrometer**
A detailed description of the instrumental characteristics of SPIRALE and of its operating mode can
be found in Moreau et al. (2005). Six tunable laser diodes emitting in spectral micro-windows (< 1 cm$^{-1}$
) in the mid-infrared domain (1250 to 3000 cm$^{-1}$) are used for in situ measurements of trace gas species
from the upper troposphere to the stratosphere. The six laser beams are injected into a multipass Heriott
cell, comprising two mirrors spaced 3.50 m apart by a telescopic mast, allowing for 434.0 m optical
path. This cell is deployed under the gondola during the flight, above ~9 km altitude, i.e. when pressure
is below ~300 hPa and thus absorption lines are significantly narrower than the scanned micro-windows.
Species concentrations are retrieved from direct absorption, by fitting experimental spectra with spectra
calculated using HITRAN 2012 database (Rothman et al., 2013) and the temperature and pressure
measured on board the gondola. Measurements of pressure (by two calibrated and temperature-regulated
capacitance manometers) and temperature (by two probes made of resistive platinum wire) allow for
conversion of the species concentrations to volume mixing ratios. Uncertainties on these parameters are
negligible regarding the other uncertainties discussed below. The instrument provides measurements
each 1.1 s, thus with a vertical resolution of a few meters depending on the vertical velocity of the
balloon (2 to 5 m s$^{-1}$). Absorption lines in the micro-windows 1260.95-1261.25 cm$^{-1}$, 1598.45-1598.85
cm$^{-1}$ and 1701.50-1701.80 cm$^{-1}$ were selected for $N_2O$, $NO_2$ and $HNO_3$, respectively. The overall
uncertainties for the volume mixing ratios have been assessed by taking into account the random errors
and the systematic errors, and combining them as the square root of their quadratic sum (Moreau et al.,
2005). There are two important sources of random errors: (1) the fluctuations of the laser background
emission signal and (2) the signal-to-noise ratio. These error sources are the main contributions for $NO_2$
giving a total uncertainty of 30% at the lower altitudes (around 15 km), gradually reduced to 20% around
20 km, and decreasing to 5% at higher altitudes (above 30 km). For $HNO_3$ these random errors are less
significant but two sources of systematic errors have to be considered: the laser line width (an intrinsic
characteristic of the laser diode) and the non-linearity of the detectors resulting in an uncertainty of 20%
on the whole profile. Concerning $N_2O$ and ozone, which are abundant and measured using detection
systems with proper linearity of the photovoltaic conversion, the overall uncertainties are 3% over the
whole vertical profile, and decrease from 10% at 14 km (i.e. for mixing ratios below 1 ppmv) to 5%
above 17 km, respectively. With respect to the above errors, systematic errors on spectroscopic data
(essentially molecular line strength and pressure broadening coefficients) are considered to be negligible
for these well studied species (Rothman et al., 2013). SPIRALE has been used routinely during the



2000's, in particular as part of European projects and satellite validation campaigns (Grossel et al., 2010; Mébarki et al., 2010; Krysztofiak et al., 2012 and 2015, and references therein).

### A.3 The DOAS remote-sensing UV-visible spectrometer

Direct solar spectra from two UV/visible DOAS spectrometers are collected onboard the azimuth-controlled LPMA/DOAS (Limb Profile Monitor of the Atmosphere/Differential Optical Absorption Spectroscopy) balloon payload which carries a sun-tracker (Hawat et al., 1995). The solar reference spectrum is usually the spectrum for which the air mass along the line-of-sight and the residual trace gas absorption are minimal. The residual absorption in the solar reference is determined using Langley's extrapolation to zero air mass. Rayleigh and Mie scattering are accounted for by including a third order polynomial in the fitting procedure. The relative wavelength alignment of the absorption cross sections and the solar reference spectrum is fixed and only the measured spectrum is allowed to shift and stretch. $O_3$ Slant Column Densities (SCDs) are retrieved from the differential structures in the Chappuis absorption band between 545 nm and 615 nm. The line-of-sight absorptions of $NO_2$ are inferred from the 435 nm to 485 nm wavelength range. Two $O_3$ absorption cross sections recorded in the laboratory at 230K and 244 K, aligned to cross sections from Voigt et al. (2001), are orthogonalized and fitted simultaneously. Broad band spectral features are represented by a fourth order polynomial. Additional complications arise from the temperature dependence of the $NO_2$ absorption cross section (Pfeilsticker et al., 1999). The $NO_2$ analysis is performed using absorption cross sections recorded in the laboratory, scaled and aligned to convolved and orthogonalized cross sections from Harder et al. (1997) taken at 217 K, and 230 K. The error bars of the retrieved SCDs are estimated via Gaussian error propagation mainly from the statistical error given by the fitting routine, the error in determining the residual absorber amount in the solar reference spectrum and the errors of the absorption cross sections. In total, typical accuracies of the DOAS $O_3$ and $NO_2$ measurements are better than 5% and 10%, respectively. The retrieval process for $NO_2$ is described in Butz et al. (2006).

Bromine monoxide (BrO) is detected in the UV wavelength range from 346 nm to 360 nm as recommended by Aliwell et al. (2002). This wavelength range contains the UV vibration absorption bands (4−0 at 354.7 nm, and 5−0 at 348.8 nm) of the $A(^2\pi) \leftarrow X(^2\pi)$ electronic transition of BrO. Typical optical densities are $10^{-4}–10^{-3}$ for UV vibration absorption bands. The set of reference spectra used contains a $NO_2$ reference spectrum for T=233 K, and two $O_3$ spectra at T=197 K and T=253 K, in order to account for temperature effects. All $NO_2$ and $O_3$ spectra were recorded with the balloon spectrograph in the laboratory. The BrO reference is the absolute cross-section measured by Wahner et al. (1988), with the wavelength calibration taken from our own laboratory measurements. Profile information was obtained by a least-squares profile inversion technique (Maximum A Posteriori) (Rodgers, 2000). Further details on the BrO DOAS-retrieval and the profile inversion can be found in Harder et al. (1998) and (2000), Aliwell et al. (2002), Dorf et al. (2006b) and Kreycy et al. (2013).

In our study we use the DOAS profile recorded in the stratosphere during the balloon ascent on 7 September 2009 between 15:15 UT (17:15 local time) and 16:35 UT (18:35 local time), corresponding to altitudes of 10 km and 30 km respectively.

### A.4 The SALOMON remote-sensing UV-visible spectrometer

The data presented in this study were obtained using a SAOZ-type UV-visible spectrometer (Pommereau and Piquard, 1994) connected to a sun/moon tracker for the detection of ozone and $NO_2$ amounts. The one-band spectral window of SALOMON between 400 and 950 nm is adequate for the retrieval of absorption features over large spectral ranges, i.e. roughly from 400 to 680 nm for ozone and from 400 to 550 nm for $NO_2$. The spectrum recorded at float altitude (more than 36.5 km) corresponds to a minimum air mass and is considered as a reference spectrum. Occultation spectra recorded for elevation angles between 0° and -5° below the gondola horizon are taken into account for the retrieval of the SCDs. Owing to the thermal insulation of the spectrometer, no spectral drift of the Fraunhofer lines and no instrumental resolution changes have been observed between the reference and the occultation spectra. The Rayleigh scattering contribution is calculated and removed from the spectra using these profiles and the spectral cross-sections given by Bucholtz (1995). Then, $O_3$ and $NO_2$ SCDs are determined by least-squares fits using the University of Bremen high resolution absorption cross-



sections convolved to the spectral resolution of the instrument (data available from http://www.iup.uni-
bremen.de/gruppen/molspec/databases/index.html). Aerosols are a major low frequency spectral
contribution which is removed by a high-pass filter to derive the $NO_2$ SCDs. All lines of sight are not
used to derive SCDs since the retrieval is performed only when signal-to-noise ratios (computed in our
case by the ratio of the fit maximum amplitude to the standard deviation between the measurement and
the fit) are greater than 1. $NO_2$ fitting errors are typically of 5-9% for SCDs crossing the altitude levels
of the volcanic aerosol layer (i.e. below ~19 km). Vertical concentration profiles have been derived
using an a posteriori least-squares inversion technique (Rodgers et al., 2000) taking into account the
fitting error and the uncertainties of the cross sections. Note that the data reduction method used in this
study is described by Renard et al. (2000) and Berthet et al. (2002).
For the flight presented in this study we have added a HR4000 UV spectrometer from Ocean Optics
to detect BrO absorption lines in the 346-360 nm range as done for the DOAS instrument. The
spectrometer is thermally insulated and regulated using Peltier devices to avoid spectral shifts. It has its
own connection to the sun tracker but collects the sunlight simultaneously with a Jobin-Yvon UV-visible
spectrometer. We use the same data reduction method as for DOAS as described in details by Dorf et
al. (2006b) to retrieve SCDs and the vertical profile of BrO. In our case the Wahner et al. BrO and
Bremen ozone and $NO_2$ cross sections are convolved to the resolution of the instrument determined in
the laboratory using a UV lamp. SCD data are smoothed to increase the signal-to-noise ratio. The altitude
grid for profile inversion is 2 km. Associated random errors are those provided by the spectral fit. The
major systematic error comes from the uncertain estimation of the residual BrO column above float
altitude.

## Appendix B: Model description

The REPROBUS 3D CTM computes the evolution of 55 species by means of about 160 photolytic
gas-phase and heterogeneous reactions, with a time step of 15 minutes in this study. A semi-Lagrangian
code transports 40 species or chemical families, typically long-lived tracers but also more unstable
compounds (Lefèvre et al., 1994; Lefèvre et al., 1998).
Temperature, winds and surface pressure are specified from the 3D European Centre for Medium-
Range Weather Forecast (ECMWF) meteorological data from the surface up to 0.01 hPa (i.e. about 80
km in altitude) on 91 levels. This results in a vertical resolution of about 0.45 km in the lower
stratosphere. REPROBUS is driven by 3-hourly ECMWF wind fields obtained by interleaving
operational analysis and forecasts because in this way spurious calculation of transport is reduced in
comparison with simulations based on 6-hourly analysis (Legras et al., 2005; Berthet et al., 2006).
Gas-phase kinetics parameters used in the present study are based on the recommendation by the Jet-
Propulsion-Laboratory (JPL) described in Sander et al. (2011). In particular for nitrogen gas-phase
chemistry, revised kinetic data were recommended because, following a number of studies (e.g. Brown
et al., 1999; Gao et al., 1999; Jucks et al., 1999; Osterman et al., 1999; Kondo et al., 2000; Prasad, 2003),
a lower rate for the reaction of $NO_2$ with OH and a higher rate for $HNO_3$ with OH significantly reduced
model-measurement discrepancies highlighted in former published work (e.g. Fahey et al., 1993; Kondo
et al., 1997; Sen et al., 1998).
The heterogeneous chemistry module includes reactions on liquid aerosols. An analytical expression
is used to calculate the equilibrium composition and volume of the $H_2SO_4$-$H_2O$ droplets as a function of
temperature and the total amounts of $H_2O$ and $H_2SO_4$ (Carslaw et al., 1995). The routine calculates the
aqueous phase concentrations for the soluble species HCl, HBr, HOCl, and HOBr to calculate the rates
of the heterogeneous reactions involving these compounds on stratospheric liquid aerosols. Reactions
of $N_2O_5$, $ClONO_2$, and $BrONO_2$ on/in sulfuric acid are usually dependent on the species' Henry's law
solubility and liquid phase diffusion coefficient in the liquid as well as the surface and/or liquid phase
reaction rates (Hanson et al., 1994; Shi et al., 2001; Sander et al., 2011). $N_2O_5$ hydrolysis takes place at
the surface of the particles (Hanson et al., 1994). As in a number of previous studies (e.g. Mills et al.,
1993; Gao et al., 1999; Bracher et al., 2005) REPROBUS computes a $\gamma$ reaction efficiency of 0.1 as
default value (0.05-0.2 in Sander et al., 2011) and which is independent of temperature and acid
composition. The reaction rate is proportional to $\gamma$ and increases with aerosol SAD. For heterogeneous





reactions involving $ClONO_2$, kinetics are taken from the well-detailed uptake model of Shi et al. (2001)
which uses the parameterization of $H_2SO_4/H_2O$ composition of Tabazadeh et al. (1997). These processes
are strongly functions of the acid composition and temperature. Note that the $\gamma$ reaction efficiency for
$ClONO_2$ described in the JPL recommendation of Sander et al. (2011) is taken from Shi et al. (2001).
The $BrONO_2$ reactivity on sulfuric acid particles is computed from the JPL parameterization which is
based on the work of Hanson (2003) and shows a rather limited dependence on acid composition and
temperature.
Initialized amounts of species are taken from a 2D model long-term simulation (Bekki and Pyle,
1994). Initialization of stratospheric chlorine precursors is based on scenarios defined by the World
Meteorological Organization (WMO, 2014). Total inorganic chlorine ($Cl_y$ = HCl + $ClONO_2$ + HOCl +
ClO + $Cl_2O_2$) is calculated by the model, and approaches 3.3 ppbv in the upper stratosphere in 2009, in
accordance with the WMO (2014). Note that as expected this value is reduced compared to the study
(3.7 ppbv) by Berthet et al. (2005). Total stratospheric inorganic bromine takes into account the
contributions from Halons, methyl bromide and very-short-lived bromine compounds to reach 19.5 pptv,
matching the scenario given by WMO (2010) updated from Dorf et al. (2006a). The 2D model
climatology (Bekki and Pyle, 1994) also provides the initialization of $H_2SO_4$ mixing ratios for the
background aerosols. Liquid particles are formed in equilibrium and are assumed to have a predefined
number density. Mean particle radii and SADs of the liquid aerosols are calculated from the number
density and the amount of $H_2SO_4$ and $H_2O$ assuming a lognormal unimodal distribution with a fixed
distribution width.



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



**Table 1**: Averaged percentage differences between model calculations and balloon-borne observations
(with respect to the measured profile) below 19 km. The model outputs from the one-dimensional
version are provided for the SPIRALE flights (see text).

| Balloon flight | Observed species | Ref-sim 3D version (no volcanic aerosols) | Bal-sim 3D version (aerosols from balloon data) | Sat-sim 3D version (aerosols from satellite data) | sim 1D version (no volcanic aerosols) | Bal-sim 1D version (aerosols from balloon data) |
|---|---|---|---|---|---|---|
| SPIRALE 7 August 2009 around 02H15 UT | $NO_2$ | +57% | +3±20% | +7% | +54% | +20% |
| | $HNO_3$ | -3% | +8±4% | +8% | -5% | +3% |
| | $NO_2/HNO_3$ | +62% | -3±19% | -1% | +60% | +16% |
| SPIRALE 24 August 2009 around 21H15 UT | $NO_2$ | +56% | -16±20% | -5% | +42% | -7% |
| | $HNO_3$ | -4% | +8±3% | +6% | -4% | +6% |
| | $NO_2/HNO_3$ | +63% | -22±20% | -10% | +48% | -12% |
| SALOMON 25 August 2009 around 19H30 UT | $NO_2$ | +51% | -16±17% | -5% | - | - |
| | $BrO$ | -66% | -34±12% | -47% | - | - |
| DOAS 7 September 2009 around 15H45 UT | $NO_2$ | +75% | -12±16% | -7% | - | - |
| | $BrO$ | -20% | 12±6% | +9% | - | - |






**Table 2**: Averaged percentage differences between the aerosol-constrained and the reference
simulations (with respect to the reference) at the dates of the various balloon flights. Differences are
provided for 3D and 1D REPROBUS simulations. Calculations are done below 19 km where
observations are available, i.e. for the same levels as in **Table 1**.

| Date | Species | Bal-sim (3D) - Ref-sim (3D) | Sat-sim (3D) - Ref-sim (3D) | Ref-sim (1D) - Ref-sim (3D) | Bal-sim (1D) - Ref-sim (3D) | Bal-sim (1D) - Ref-sim (1D) |
|---|---|---|---|---|---|---|
| 7 August 2009 around 02H15 UT | $NO_2$ | -34±13% | -31% | -3% | -24% | -22% |
| | $HNO_3$ | +10±4% | +9% | +7% | +15% | +8% |
| | $NO_2/HNO_3$ | -40±13% | -36% | -9% | -34% | -27% |
| 24 August 2009 around 21H15 UT | $NO_2$ | -46±12% | -39% | -9% | -41% | -34% |
| | $HNO_3$ | +13±3% | +11% | +2% | +13% | +11% |
| | $NO_2/HNO_3$ | -52±12% | -44% | -10% | -47% | -41% |
| 25 August 2009 around 19H30 UT | $NO_2$ | -47±10% | -40% | - | - | - |
| | BrO | 181±74% | 128% | - | - | - |
| 7 September 2009 around 15H45 UT | $NO_2$ | -50±8% | -47% | - | - | - |
| | BrO | 38±8% | 34% | - | - | - |





**Table 3**: Simulated changes on various stratospheric key species due to the Sarychev volcanic aerosols
over the August-September 2009 period at 16.5 km. Calculations are done from the Sat-sim
simulation. Effects for daytime and nighttime conditions are provided depending on statistically
significant amounts in the diurnal cycle of a given compound. The contribution of $BrONO_2$ hydrolysis
to changes on the various species is also shown (see text).

| Species | All chemistry | | | | $BrONO_2$ hydrolysis effect | |
|---|---|---|---|---|---|---|
| | 12H UT | | 00H UT | | 12H UT | 00H UT |
| $NO_x$ | -0.23 ppbv | -44% | -0.19 ppbv | -48% | 1.8% | 1.1% |
| $NO_2$ | -0.12 ppbv | -43% | -0.19 ppbv | -48% | 1.8% | 1.1% |
| $NO$ | -0.11 ppbv | -45% | --- | --- | 2.0% | --- |
| $HNO_3$ | +0.31 ppbv | +11% | +0.31 ppbv | +11% | -2.3% | -0.9% |
| $N_2O_5$ | -0.08 ppbv | -80% | -0.12 ppbv | -66% | -3.6% | -3.1% |
| $ClONO_2$ | +0.02 ppbv | +16% | +0.02 ppbv | +22% | 66.2% | 60.6% |
| $HCl$ | -0.02 ppbv | -3% | -0.02 ppbv | -3% | 58.8% | 58.9% |
| $ClO_x$ | +5.77 pptv | +106% | --- | --- | 39.3% | --- |
| $ClO$ | +5.77 pptv | +106% | --- | --- | 39.3% | --- |
| $HOCl$ | +2.17 pptv | +217% | +1.16 pptv | +346% | 47.4% | 50.1% |
| $BrONO_2$ | -1.37 pptv | -33% | -4.15 pptv | -70% | 18.3% | 98% |
| $BrO$ | +0.94 pptv | +22% | --- | --- | 16.2% | --- |
| $HOBr$ | --- | --- | +3.89 pptv | +141% | --- | 98.8% |
| $HO_x$ | +1.41 pptv | +51% | --- | --- | 24.1% | --- |
| $OH$ | +0.05 pptv | +16% | --- | --- | 44.1% | --- |
| $HO_2$ | +1.36 pptv | +56% | --- | --- | 23.1% | --- |
| $O_3$ | -13.1 ppbv | -1.1% | -12.6 ppbv | -1.1% | 22.5% | 26.3% |


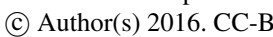



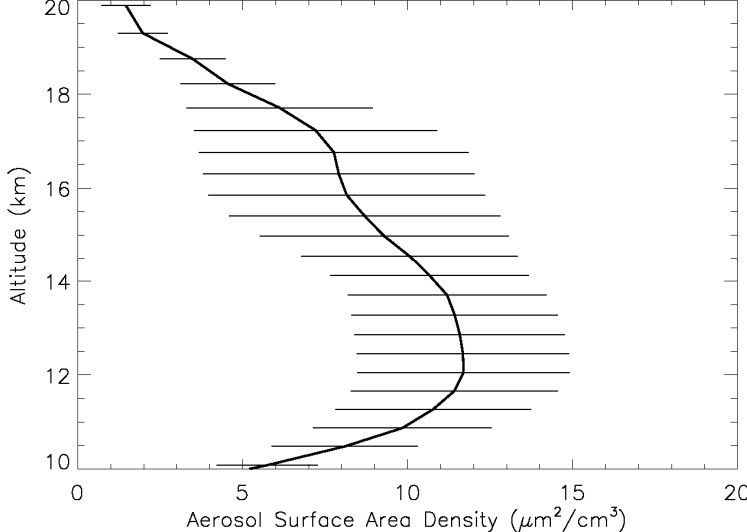

**Figure 1:** Range of aerosol SAD values derived from several balloon-borne observations in the lower
stratosphere in summer 2009. Based on Table 1 in the work of Jégou et al. (2013) this average profile
excludes data supposed to be spoiled by balloon outgassing as revealed from simultaneous in situ
water vapour observations.





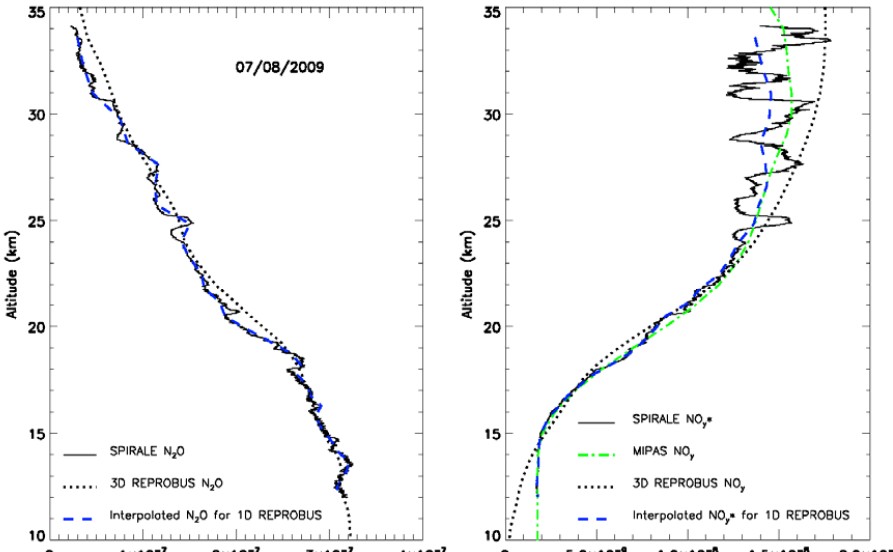

**Figure 2:** In situ vertical profiles recorded by the SPIRALE instrument used to constrain 1D
simulations of the REPROBUS model. Left: profile of $N_2O$ recorded on 7 August 2009 (black line)
compared to the results from the 3D version of REPROBUS (dotted line). Right: profile of $NO_y$
inferred from $N_2O$ observations converted using the $N_2O$-$NO_y$ correlation curve presented in **Figure 3**
(referred to as $NO_y$*). Also shown are the $NO_y$ profiles from the 3D version of REPROBUS (dotted
line) and the MIPAS averaged data (green line). The 1D version of REPROBUS is computed with the
profiles interpolated to the model resolution (blue lines).



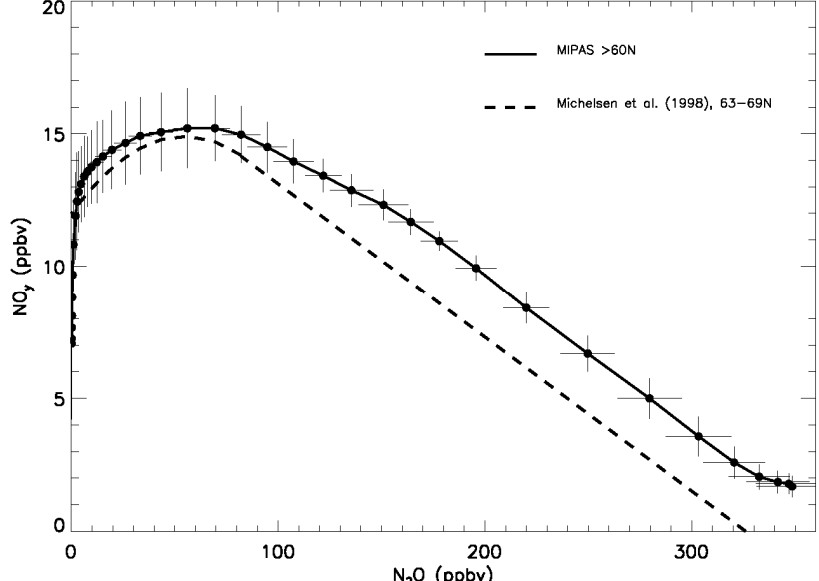

**Figure 3:** $N_2O$-$NO_y$ correlation curve inferred from IMK/IAA V5R_220 MIPAS-Envisat data at high
latitudes (> 60°N) in July-August 2009 (full line). Error bars reflect the spread of the data. The former
Michelsen et al. (1998) correlation is also shown for comparison (dashed line).





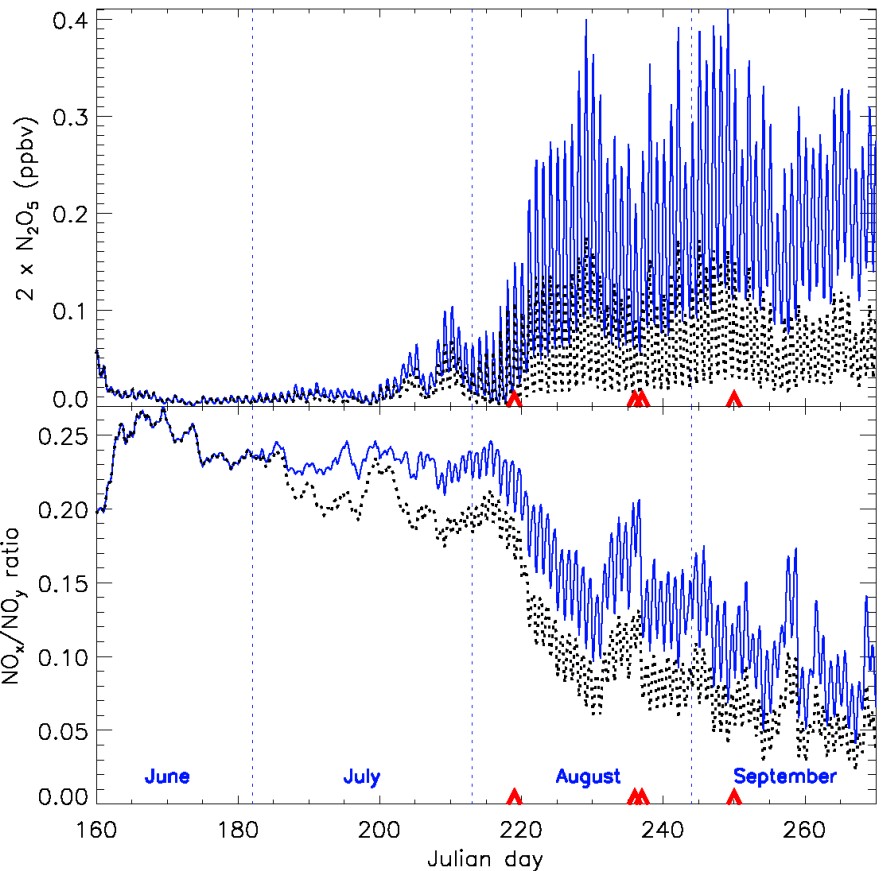

**Figure 4**: Seasonal variation of $N_2O_5$ (a) and of the $NO_x/NO_y$ ratio (b) simulated by the REPROBUS
CTM above Kiruna in Northern Sweden (67.5°N, 21.0°E) around 17.5 km. The simulation driven by
non-volcanic aerosol contents (Ref-sim) is shown in blue. The black dotted line is the REPROBUS
simulation driven by volcanic aerosol levels from STAC balloon-borne observations (Bal-sim). Red
triangles represent the dates of the balloon flights. $N_2O_5$ recovery is onset at the beginning of August
(day 213 is August 1, 2009) i.e. when SZA become >90°.



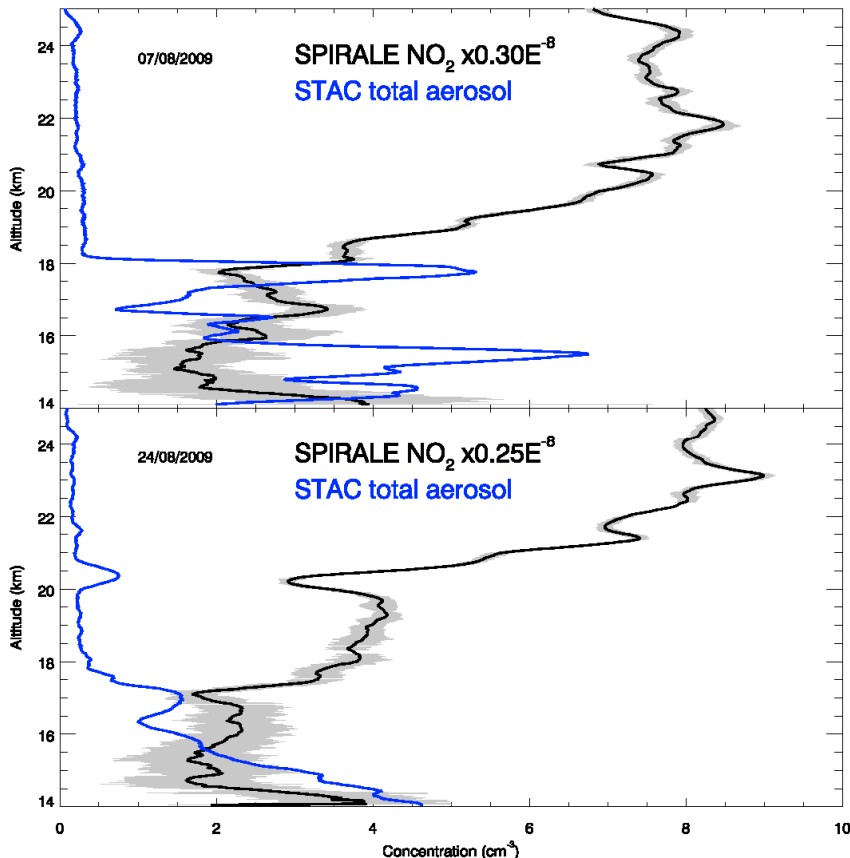

**Figure 5:** Vertical profiles of NO₂ observed by the SPIRALE balloon-borne instrument (black line) on
7 and 24 August 2009 compared to the total aerosol concentration profiles simultaneously recorded by
the STAC aerosol counter (blue line) above Kiruna during balloon ascent. SPIRALE data have been
averaged over 250 m (corresponding to ~1 minute of measurements).



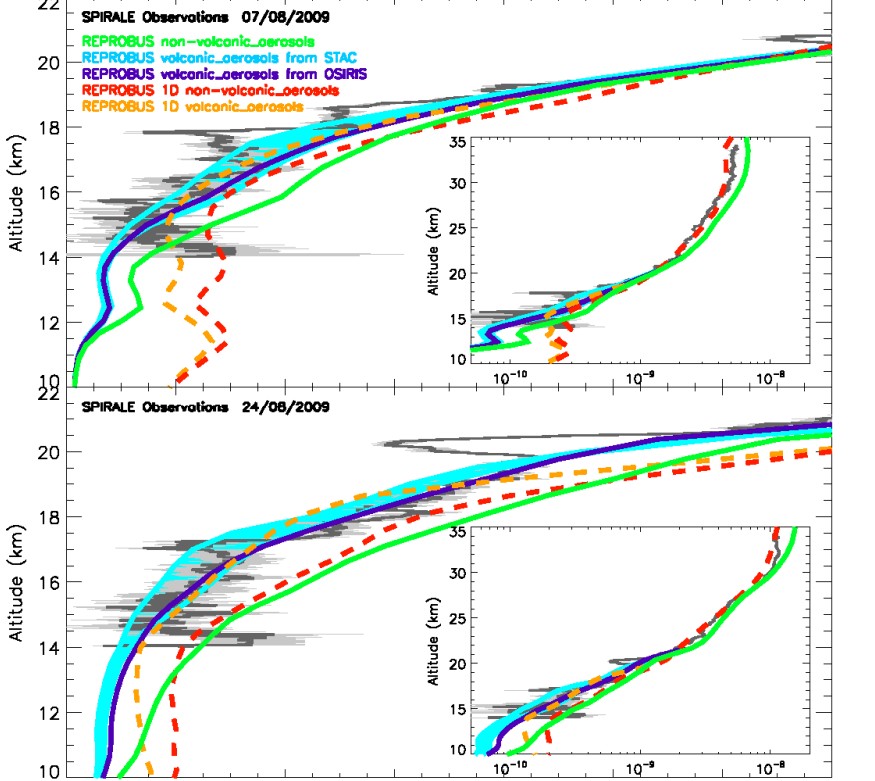

**Figure 6:** Vertical profile of $NO_2$ observed by the SPIRALE balloon-borne instrument (black line)
above Kiruna during balloon ascent between 02:00 and 02:30 UT for the 7 August 2009 flight (top)
and between 21:00 and 21:30 UT for the 24 August 2009 flight (bottom). Model outputs (available
every 15 minutes) are provided for the closest location of the instrument and interpolated to the time
of observations. Three-dimensional simulations have been driven without volcanic aerosols (green),
with volcanic aerosols from balloon-borne observations (blue shaded area) and with volcanic aerosols
from satellite data (dark blue line). Results from a one-dimensional (1D) version of the REPROBUS
model (dashed lines) computed using hybrid $NO_y$ profiles ($NO_y$*) derived from the observed profiles
of $N_2O$ are also provided (see text), with in red the non-volcanic reference simulations and in orange
the calculations driven with volcanic aerosols from the mean observed balloon-borne profile presented
in **Figure 1**.



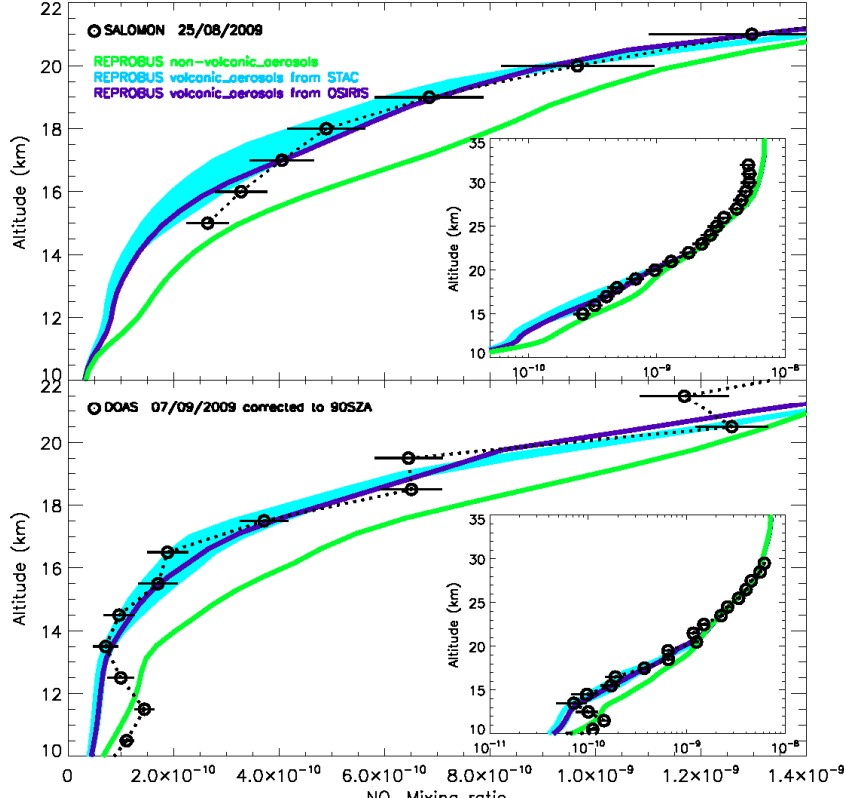

**Figure 7:** (top) Vertical profile of NO$_2$ recorded by the SALOMON instrument (black lines) obtained
during solar occultation between 18:50 and 19:30 UT on 25 August 2009 above Kiruna. Chemistry-
transport model simulations computed with no volcanic aerosols (green line), with volcanic aerosols
from balloon-borne observations (blue shaded area) and with volcanic aerosols from satellite data
(dark blue line) are shown. The model output is provided for the closest location of the tangent points.
(bottom) Vertical profile of NO$_2$ recorded by the DOAS instrument (black lines) on 7 September 2009
above Kiruna. The DOAS profile has been recorded during the balloon ascent and has been converted
to 90°SZA (~17:30 UT) as well as the simulated profile.




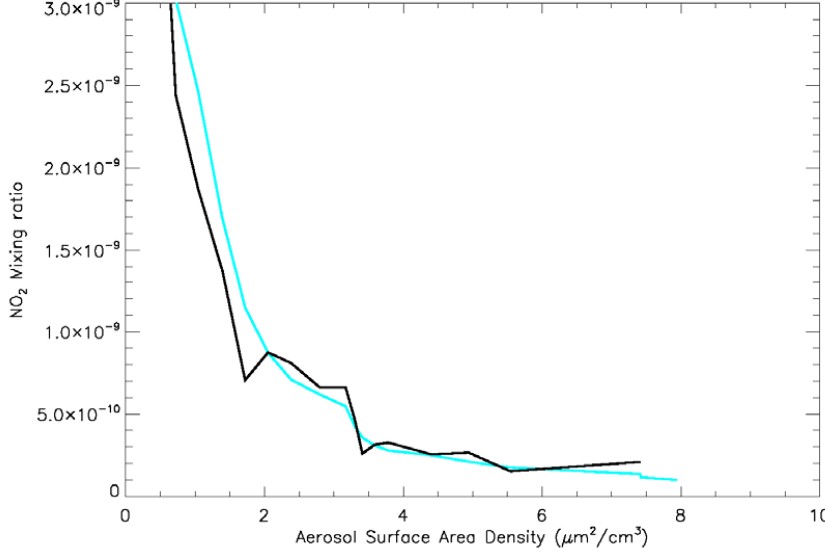

**Figure 8**: NO$_2$ mixing ratio as a function of aerosol SAD as simultaneously observed in the lower
stratosphere by the SPIRALE and STAC instruments on 24 August 2009 (black curve). The result
from the REPROBUS Bal-sim simulation is also plotted (blue curve).



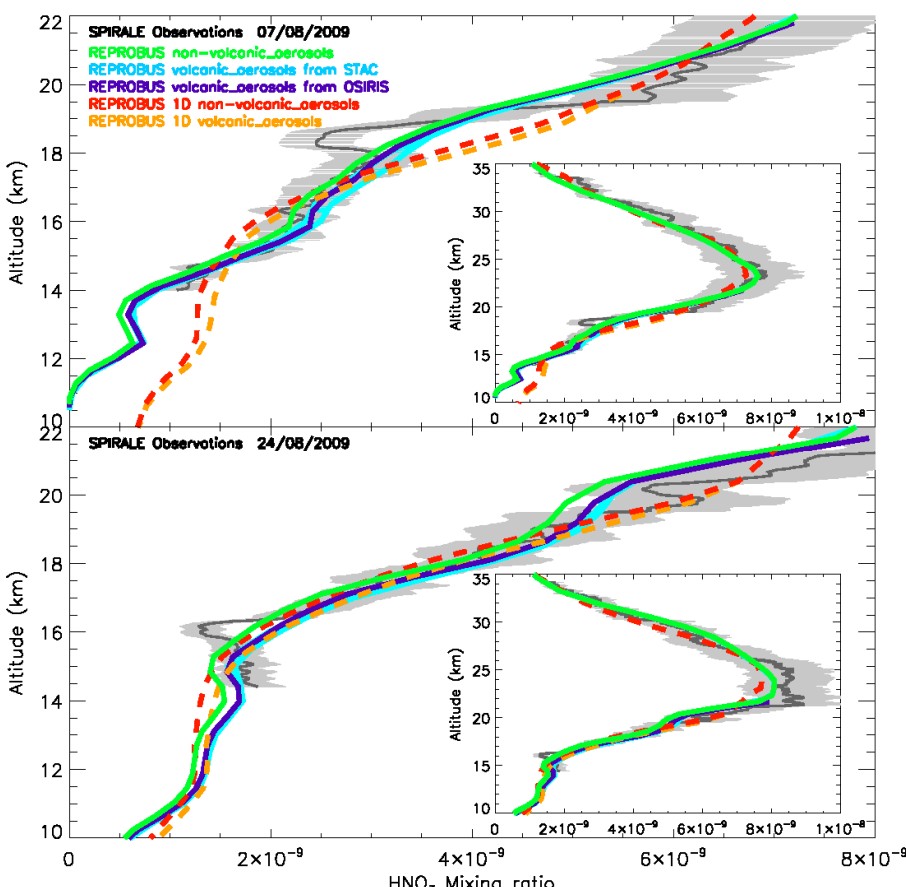

5    **Figure 9:** same as **Figure 6** but for HNO₃.



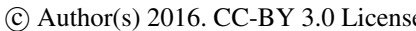

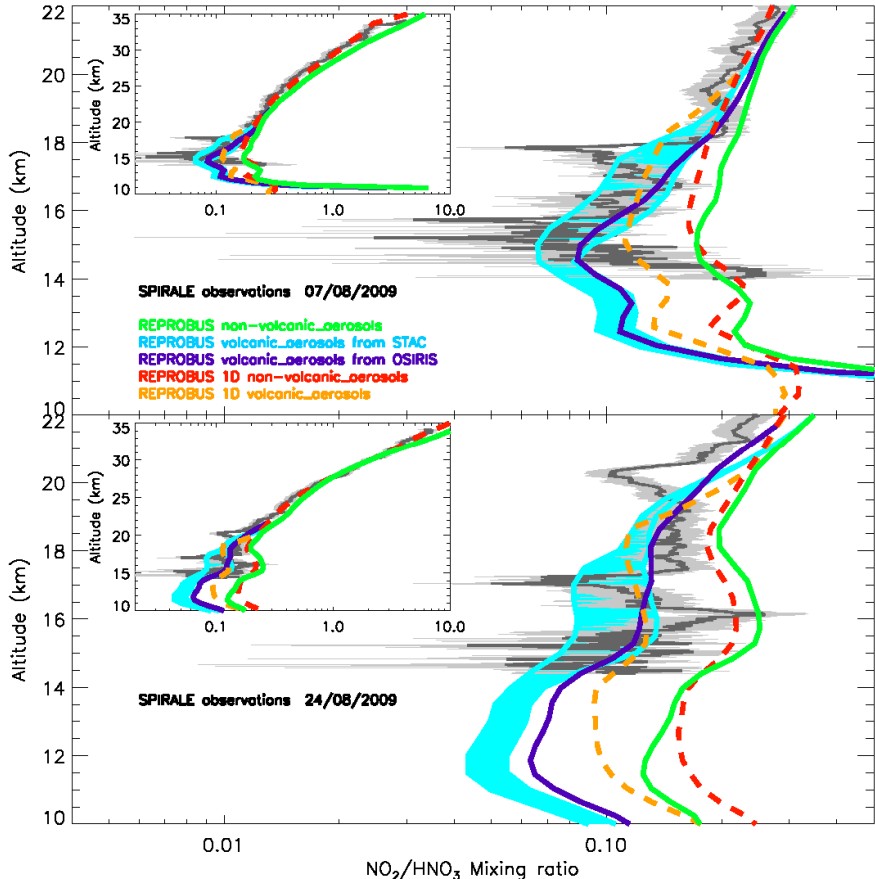

**Figure 10:** same as **Figure 8** but for the NO$_2$/HNO$_3$ ratio.



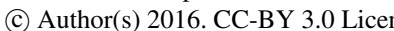

**Figure 11:** same as **Figure 7** but for BrO. The BrO profile from DOAS in the lower stratosphere has
been obtained between 15:15 and 15:55 UT.





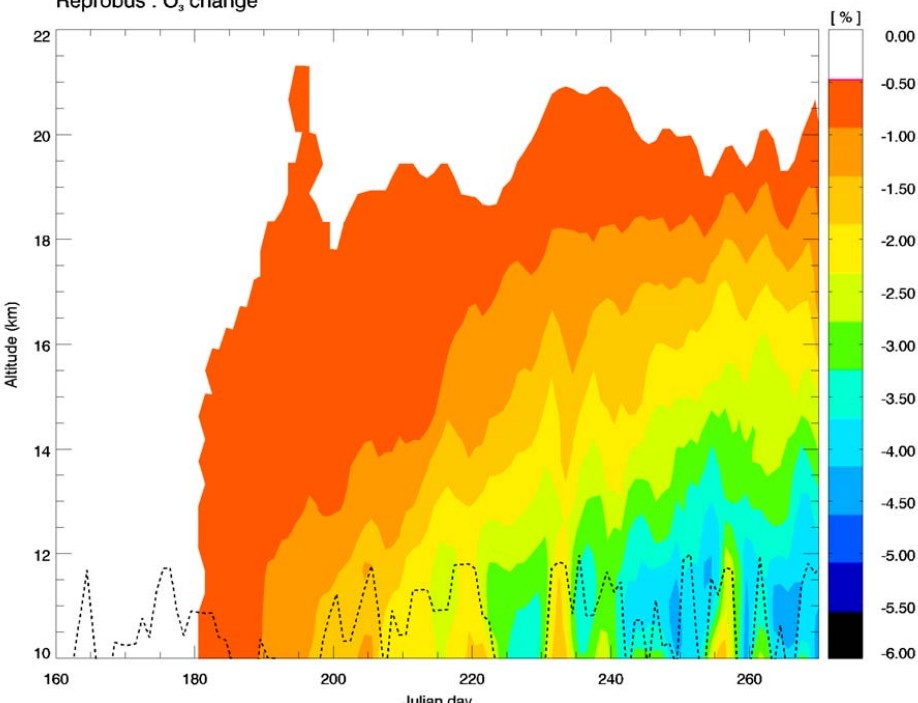

**Figure 12**: Percentage changes in ozone over Kiruna (67.5°N, 21.0°E) as a function of altitude and
time between 1 July and 1 October 2009. Calculations are done by subtracting outputs from the
volcanic simulation driven by OSIRIS observations with the background simulation. The position of
the tropopause is given by the black dotted line.