# Peer review of "Impact of a moderate volcanic eruption on chemistry in the lower"

_Atmospheric Chemistry and Physics, 2016_

## Referee Comment (RC1) · Anonymous Referee #1 · 26 Oct 2016

The main purpose of this paper is to present measurements of various NOy family species from several balloon flights through a region of volcanically enhanced aerosol surface area density, then show that the observations are consistent with chemical perturbations calculated by a 3D chemistry transport model driven with reanalysis meteorology. The paper also presents estimates of O3 depletion caused by the volcanic aerosols and discusses chemical perturbations to the inorganic Br and Cl families. The measurements confirm our existing understanding of the role of heterogeneous chemistry in NOy partitioning; this is not new science.

The paper is quite long for the content. The abstract describes what is in the paper, but it is unclear whether the results are new or significant. The Intro describes the

study as "the chemical impact of a short-term change in the amount of stratospheric sulfate aerosols resulting from one of these 'moderate' volcanic eruptions on some key aspects of stratospheric chemistry and on ozone loss", but the authors' have not shown how this advances the state of knowledge in this field, which is already well studied. Just how well studied this is, is indicated by the 136 references given and that the majority of them are dated before 2000. The authors may not be aware of relevant recent results. For example, they state that most models used to estimate chemical effects of aerosols are 2D models (p. 15, line 50) but this was true 10 years ago. Recent results using 3D models are overlooked (e.g., a CCM study by Aquila et al, JAS 2013, and a CTM study by Dhomse et al., GRL 2015). These recent papers also confirm our understanding of the role of volcanic aerosols in NOy partitioning by showing good model agreement with observations. This underscores my concern that there is not new science in this manuscript.

The style is verbose and the writing can be confusing. Here is an example (starting line 47, p. 4). We are told that solar zenith angle impacts the retrieved profile so it needs correction with a photochemical model, but then we are told that using such a model would introduce larger errors in the retrieval (so it's a bad idea to correct?). In the next paragraph they estimate the correction anyway, saying it is only 3%. But then they cite the correction as being a 24% effect on a particular balloon flight. I don't know what to conclude here, there is no clear message. A paragraph on p. 13 gives a quantitative estimate of the impact of aerosols on O3 depletion using a simulation with varying amounts of aerosols. At the end of the paragraph we are told not to take the results too seriously because the model is missing (presumably) relevant chemical reactions. These two examples illustrate a common problem with the manuscript: a meandering discussion without a clear message. The manuscript, not counting tables, figures, and captions, is more than 10,000 words. This is too long for the presentation of a few balloon profiles and model simulations that show aerosol impacts. The information in Tables 1 and 2 shows percentage disagreements between simulations and observations. This is unnecessary and corresponding figures that show model/data

comparisons are sufficient.

This paper, as is, is not ready for publication. The amount of new material/new science is small and I recommend a much shorter, more concise presentation of the observations and comparisons to simulations. Something half the current length might be appropriate.

———————————————————

---

## Referee Comment (RC2) · Anonymous Referee #2 · 28 Nov 2016

**Review of:**

**Impact of a moderate volcanic eruption on chemistry in the lower stratosphere: balloon-borne observations and model calculations**

by G. Berthet et al.

**Anonymous Reviewer 2**

**1   General Comments**

The manuscript presents a measurement model intercomparison study of the effects of the aerosol plume of the Sarychev volcano eruption in June 2009 on the high latitude lower stratospheric nitrogen, halogen and HOx chemistry and the relevance to the lower stratosphere ozone budget. The study employs a broad spectrum of balloon-borne measurements of aerosols and trace-gases as well as several sets of satellite aerosol data. The model employed is the REPROBUS 3D CTM that is well-known from several studies mainly focussing on process understanding. Here the focus is on the effects of the plume of a moderate volcanic eruption (as opposed to several studies on major volcanic plumes as Pinatubo) which is worthwhile since a series of these moderate eruptions is supposed to be one cause of the long-term increase in the global stratospheric aerosol density. Several interesting aspects of the chemistry are presented and special features of the combination of this moderate eruption plume with high-latitude lower stratospheric chemistry are identified. Therefore a general interest for such a study is clearly given and ACP represents a well-suited platform for the publication.

However, the reasoning for exploring the effect of a moderate eruption on chemical composition at Arctic latitudes in summer must be better rationalized in the introduction. Why is this important? Currently only the relevant chemical, aerosol and some dynamical processes are explained.

From the satellite data presented in Haywood (2010) and Jegou (2013) and also Fig.5 it seems obvious that the plume is not homogeneously mixed over the Arctic region by August/September 2009. Therefore the expected horizontal and vertical structure has to be discussed in some more detail. The STAC balloon data from different flights needs to be introduced in Fig. 1 (e.g. grey underlayed traces?), not just

ranges of the observations so the reader gets a better impression of the vertical structure of the plume and its variability. Horiz. and vertical variability should be discussed at least. Currently no filtering for high/low or even background aerosol regions is done for the interpretation of the data which may well be warranted but must be better supported.

In order to explore the sensitivities of the model study to different parameters such as differences in aerosol surface area or dynamical effects a number of differently constrained simulations have been carried out and are intercompared in the figures and tables presented. Therefore partly the figures and tables and consequently the discussion gets quite busy and confusing. The results for the runs termed sat-sim and bal-sim generally don't differ by more than 10%, mostly much less. Therefore this just needs to be shown in one plot (Fig. 6) but then can be neglected just stating that differences for other species are also minor. The same is true for the 1D simulations which are meant to check on dynamical influences. Once the results of these model experiments have been stated the following discussions can be simplified a lot by leaving all the other simulations out. Especially Table2 should be considerably simplified, I'm not aware that all the various differences presented there are even discussed in the text.

The balloon-borne measureemnts aquired during the Strapolete experiment represents a data set that nicely covers an interesting episode of aerosol enhancement in this atmospheric domain and therefore publication is of it's own value. Possibly a link or links to the appropriate data base(s) should be also given in order to enable use of the data for other studies.

In general I think the manuscript should be published, however, after considering the general and detailed comments and streamlining the presentation and discussion.

**2 Detailed and minor Comments**

**p4:l16** Is the wording "We focus here on ..." meant to discriminate against the aerosol measurements or does it just refer to gas-phase data?

**p4:l16ff** Why are there only ascent data used for the SPIRALE measurements? It would be interesting to see also descent data to get a feeling on variability and possible contamination issues on ascent since the cell extends below the payload.

**p5:l3ff** This paragraph is somewhat chaotic and hard to understand and should be polished. Before switching to the BrO profiles a new paragraph might be started.

**p5:l21ff** The fact that REPROBUS is used without any detailed sulfur chemistry should be clearly stated right away then referring in which different ways the aerosol plume is prescribed.

**p5:l53** The Haywood (2010) reference is missing.

**p6:l33** The analytical expression for the derived correlation in Fig.3 should be given.

**p6:l37** Better leave out the phrase *in the model*.

**p9:Sect.3.4** can be considerably shortened since it doesn't add new results (see Fig.9).

**p9:l53** Why is the nomenclature changed here to Balloon-aero-sim instead of Bal-sim etc.?

**p10:Sect.3.6** The stated improvements of the 1D simulations above 20km are not at all obvious to me and are certainly not significant improvements that can be employed for the conclusion drawn in this section.

**p12:l10ff** With the introduction of Fig. 11 the dramatic difference of the two BrO profiles should be explained (sza?). Also what is the tropopause height for the DOAS measurement? More than 1 ppt of BrO below the tropopause seems quite suspicious to me.

**p12:l35** The meaning of the percentage changes for switching off the BrONO2 hydrolysis must be more clearly explained. For the example given it should be 16% of the daytime BrO production not 18%.

**p12:l39** This result implies ...

**p12:l52** ... the active chlorine family species ...

**p13:l30ff** I propose to use absolute values for the ozone loss discussion (see the comment on Fig.12).

**p13:l39** ... into the lower stratosphere.

**p15:l33** When switching to absolute values for the accumulated ozone losses the discussion why "largest" losses occur just above the tropopause will become obsolete, I guess.

**p16:l35** It might be interesting ...

**p18:l19** Does the *uncertainty* represent accuracy? Since several balloon-borne measurements are used along with each other and are compared to other aerosol SAD data it is not sufficient to just state the precision of the measurements.

**p18:l41** I guess *overall uncertainty* represents accuracy? Rename or otherwise state the accuracy.

**p19:l17** The Voigt et al. reference is missing. Also the Pfeilsticker et al. reference (l20). Please check over completely!

**p21:l3** ... strong functions ...

**p22** All citations should be thoroughly rechecked since several citations have been missing from the references list.

**Table 2** Several of the tabulated differences are not used at all in the text. The table can be reduced considerably or even removed completely.

**Table 3** In the caption it should better read: Numbers are taken from the Sat-sim simulation. Also ... BrONO2 hydrolysis (Reaction 4) to changes ... would help.

**Fig.1** I propose to show the traces of the individual measurements on the plot to give the reader an impression on the variability. Also the altitudes shown should be extended somewhat to 25km and possibly below 10km to give a better impression on the vertical extend of the plume.

**Fig.2** For my taste in the right panel SPIRALE HNO3 should be included.

**Fig.5** The grey shaded (error bars?) on the NO2 profile should be explained in the caption.

**Fig.9** Due to the high partitioning into HNO3 this species is not sensitive for the aerosol effect. This plot therefore can be left out.

**Fig.11** The solar zenith angles of the profiles have to be given in the caption due to the diurnal variation of BrO.

**Fig.12** Showing the percentage changes in ozone is somewhat misleading. In order to point out the altitude regime of highest impact in terms of ozone loss an absolute scale should be used like loss rates of ppb/day. a percentage change of 5% doesn't have any major effect if ozone levels are negligible at the TP.

---

## Author Comment (AC1) · 7 Jan 2017

We thank the reviewer for his/her comments on the manuscript. Hereafter, please find point-by-point answers.

Reviewer's comment: "The main purpose of this paper is to present measurements of various NOy family species from several balloon flights through a region of volcanically enhanced aerosol surface area density, then show that the observations are consistent with chemical perturbations calculated by a 3D chemistry transport model driven with reanalysis meteorology. The paper also presents estimates of O3 depletion caused by the volcanic aerosols and discusses chemical perturbations to the inorganic Br and Cl families. The measurements confirm our existing understanding of the role of hetero-

geneous chemistry in NOy partitioning; this is not new science."

→ Our goal is not to discuss the general understanding of the role of heterogeneous chemistry on NOy partitioning which, we agree with the reviewer, has been widely investigated. We aim at quantifying for one of the first times (if we consider the work of Adams et al. 2016 in ACPD) the chemical impact of "moderate" volcanic eruptions (focusing here on the Sarychev eruption because high-resolution data are available, see below) which are supposed to be the main explanation (possibly together with the increasing SO2 emissions in Asia) for the increase in the stratospheric aerosol content over the past 15 years (Vernier et al., GRL, 2011). We specifically investigate this eruption because stratospheric chemistry is impacted from August to November, i.e. in extra-vortex and high temperature conditions, precluding polar winter processes. The manuscript is not only focused on effects on nitrogen chemistry but also on bromine through measurements of stratospheric BrO which are made by a very limited number of teams in the world, and rarely obtained in high aerosol loading conditions. An interesting result is that chlorine partitioning is significantly controlled by enhanced BrONO2 hydrolysis through further production of OH when ClONO2 hydrolysis is not efficient. An interest is to try provide some comparisons with the Pinatubo effects though both events are not easily comparable as a result of various features (e.g. altitude/latitude of injection) as said in the manuscript. Also, we provide for the first time high-resolution data obtained within the volcanic plume from a moderate eruption to investigate locally the chemical effects. The limited vertical extent of the aerosol plume produced by this kind of eruption in comparison with the Pinatubo event requires (rare) observations well resolving the lower stratosphere. This is not easily achieved by space-borne instruments. To summarize, we are convinced that it is new and of interest to investigate stratospheric chemistry and effects on ozone using valuable tools in a "background" aerosol loading context which is different from the very low aerosol content situation in the late 90's and early 2000 years, especially when almost no one has done it and considering that the scientific community has been using the Pinatubo major eruption as the only reference so far.

Reviewer's comment: "The paper is quite long for the content. The abstract describes what is in the paper, but it is unclear whether the results are new or significant. The Intro describes the study as "the chemical impact of a short-term change in the amount of stratospheric sulfate aerosols resulting from one of these 'moderate' volcanic eruptions on some key aspects of stratospheric chemistry and on ozone loss", but the authors' have not shown how this advances the state of knowledge in this field, which is already well studied."

→ Following our answer to the previous reviewer's comment, we have modified the abstract and introduction accordingly. We better describe the rationale of our study, i.e. focus on a mid-latitude eruption in a high-temperature stratospheric context. Finally, we can argue that our investigation of a specific moderate volcanic event initiates future quantification of the possible chemical impacts resulting from the series of eruptions since the beginning of the 21st century, especially those occurring in the tropics (with longer aerosol residence times) and/or in winter (low temperature conditions enhancing chlorine catalytic cycles).

In the abstract, we have modified the text: "...only "moderate" but recurrent volcanic eruptions have modulated the stratospheric aerosol loading and are assumed to be one cause for the reported increase in the global aerosol content over the past 15 years. This particular enhanced aerosol context raises questions about the effects on stratospheric chemistry which depend on the latitude, altitude and season of injection. In this study, we focus on the mid-latitude Sarychev volcano eruption in June 2009 which injected 0.9 Tg of sulfur dioxide (about 20 times less than Pinatubo) in a lower stratosphere mainly governed by high stratospheric temperatures. Together with in situ measurements of aerosol amounts, we analyse high-resolution in situ and/or remote-sensing observations of NO2, HNO3 and BrO from balloon-borne infrared and UV-visible spectrometers launched in Sweden in August-September 2009."

We have also added: "We show that the chlorine partitioning is significantly controlled by enhanced BrONO2 hydrolysis."

And added at the end of the abstract: "...the simulated chemical ozone loss due to the Sarychev aerosols is low with a reduction of -22 ppbv (-1.5%) of the ozone budget around 16 km. This is at least 10 times lower than the maximum ozone depletion from chemical processes (up to -20%) reported in the northern hemisphere lower stratosphere over the first year following the Pinatubo eruption. This study suggests that moderate volcanic eruptions have limited chemical effects when occurring at mid-latitudes (restricting residence times) and outside winter periods (high temperature conditions). However, among the other reported moderate eruptions it would be of interest to investigate longer lasting tropical volcanic plumes or sulfur injections in the wintertime low temperature conditions."

The end of the introduction has been changed to: "...In periods following major eruptions, the year-to-year variability in stratospheric ozone at northern mid-latitudes appears closely linked to dynamical changes induced by the volcanic aerosol radiative perturbation (e.g. Telford et al., 2009; Aquila et al., 2013) and to changes in chlorine partitioning (e.g. Solomon et al., 1999; Chipperfield, 1999). Effects on stratospheric chemistry are expected in periods of elevated chlorine levels from anthropogenic activities (Tie and Brasseur, 1995; Solomon et al., 1996)... Their effects depend on the amount of released SO2 and on latitudes and altitudes of injection which directly influence aerosol residence times. The season of the eruption is also important for photochemical processes which are directly connected to temperatures and solar illuminations. The goal of this paper is to show how such moderate eruptions are likely to modify the chemical balance of the northern hemisphere lower stratosphere at periods excluding wintertime/springtime halogen-activating photochemistry. We specifically focus on the eruption of the Sarychev volcano on 15 and 16 June 2009 which injected 0.9 Tg of sulfur dioxide in the lower stratosphere (Clarisse et al., 2012) resulting in enhanced sulfate aerosol loading up to 19 km, for a period of about 8 months ending before winter (Haywood et al., 2010; Kravitz et al., 2011; O'Neill et al., 2012; Jégou et al., 2013). The approach consists in analysing some key aspects of lower stratospheric chemistry and ozone loss in a context of high aerosol surface area densities

and high stratospheric temperatures using balloon-borne observations conducted in August-September 2009 from Kiruna/Esrange in Sweden (67.5°N, 21.0°E) within the frame of the STRAPOLETE project. To our knowledge we show here the first high-resolution in situ observations of chemical compounds obtained within the volcanic aerosol plume of a moderate eruption. We show that at the period on which the study is focused N2O5 has reformed and the role of its hydrolysis becomes important again after the sunlit summer period justifying the use of these balloon data for the investigation of heterogeneous processes. Aerosol-constrained simulations using a 3D Chemistry Transport Model (CTM) are compared to the observations. These model calculations ignore possible dynamical effects induced by the volcanic aerosols but are used to estimate the amplitude of the chemical impacts and ozone loss with some comparisons with the post-Pinatubo eruption period."

Reviewer's comment: "Just how well studied this is, is indicated by the 136 references given and that the majority of them are dated before 2000. The authors may not be aware of relevant recent results. For example, they state that most models used to estimate chemical effects of aerosols are 2D models (p. 15, line 50) but this was true 10 years ago. Recent results using 3D models are overlooked (e.g., a CCM study by Aquila et al, JAS 2013, and a CTM study by Dhomse et al., GRL 2015). These recent papers also confirm our understanding of the role of volcanic aerosols in NOy partitioning by showing good model agreement with observations. This underscores my concern that there is not new science in this manuscript."

→ The reviewer is right, 3D modelling studies of the Pinatubo eruption and its impact on stratospheric ozone have been available since the years 2000. We have removed the following sentence highlighted by the reviewer (p. 15, line 50 in the ACPD version): "We also note that former studies mostly used 2D simulations to investigate the chemical effects of the enhanced aerosol burden following the Pinatubo eruption with some limitations in terms of meridional transport simulations".

However, most of these recent studies mentioned by the reviewer focus on the dynamical mechanisms influencing ozone variability and hemispheric asymmetries in a volcanically perturbed stratosphere (and all cases, the post-Pinatubo eruption period). Though both papers cited by the reviewer present responses of NO2, and not only of ozone, to the Pinatubo aerosols they deal with a different issue than our work which is centred on detailed stratospheric chemistry allowing to estimate the halogen effect on ozone loss, specifically in the northern hemisphere.

Anyway, we have added some references dealing with 3D model calculations in the introduction, section 5.1 (together with the Telford et al. (2009) paper which is already cited) and/or in the final discussion: Al-Saadi et al. (2001), Aquila et al. (2013) and Dhomse et al. (2015).

Reviewer's comment: "The style is verbose and the writing can be confusing. Here is an example (starting line 47, p. 4). We are told that solar zenith angle impacts the retrieved profile so it needs correction with a photochemical model, but then we are told that using such a model would introduce larger errors in the retrieval (so it's a bad idea to correct?). In the next paragraph they estimate the correction anyway, saying it is only 3%. But then they cite the correction as being a 24% effect on a particular balloon flight. I don't know what to conclude here, there is no clear message."

→ We mean that a photochemical correction is not systematically used for remote sensing observations. The applicability of this correction procedure has been widely discussed in the literature as reported in the former works provided in our manuscript and might vary from one study to another because: 1) the need to apply this procedure depends on a combination of different parameters such as the considered chemical compound, the observation geometry (i.e. balloon ascent or occultation) and daytime (SZA variation) and 2) using theoretical (model) calculations to correct measurements requires good knowledge of the diurnal variation of the retrieved species and is likely to depend on the model used.

To avoid confusion in this part of the manuscript we have removed the sentence: "However, some retrievals from occultation measurements do not include corrections for diurnal variations in concentrations because such corrections are strongly dependent on the photochemical model used in the retrieval algorithm and are likely to result in additional errors (Randall et al., 2002)."

We now write: "Variations of solar zenith angle (SZA) along solar occultation lines of sight and associated concentration variations are likely to impact the retrieved vertical profiles near sunrise and sunset especially below 20 km (Newchurch et al., 1996; Ferlemann et al., 1998). Some works propose to use a photochemical model to correct for this effect (e.g. Harder et al., 2000; Butz et al., 2006) depending on the considered chemical compound, the observation geometry (i.e. balloon ascent or occultation) and daytime (SZA variation). Typically concentrations are converted to values expected at 90° SZA. In our study, the NO2 profile from the SALOMON instrument recorded on 25 August 2009 from a typical solar occultation at constant float altitude is not photochemically corrected since conversion to 90° SZA conditions results in differences of less than 6%, in agreement with the work of Payan et al. (1999). The vertical profile observed by the DOAS instrument was recorded on 7 September 2009 with a different observation geometry, i.e. during the balloon ascent. In this case applying a photochemical correction gives differences of 24% and the model-measurement comparison is done for SZA = 90°."

Reviewer's comment: "A paragraph on p. 13 gives a quantitative estimate of the impact of aerosols on O3 depletion using a simulation with varying amounts of aerosols. At the end of the paragraph we are told not to take the results too seriously because the model is missing (presumably) relevant chemical reactions. These two examples illustrate a common problem with the manuscript: a meandering discussion without a clear message."

→ About the last sentence of paragraph 5.1 (p13 line 37 in the ACPD version), we are specifically referring here to model calculations at (or very close to) the tropopause. We were mentioning that the model is driven with stratospheric chemistry and does

Interactive
comment

not account for the detailed chemistry of tropospheric organic compounds (e.g. PAN, etc.) possibly impacting the ozone budget (production in the case of PAN) at (or very close to) the tropopause level.

However, we agree that the writing might be confusing and we have removed the sentence since it becomes obsolete when absolute ozone destruction values (in ppbv) are shown as suggested by reviewer 2.

Reviewer's comment: "The manuscript, not counting tables, figures, and captions, is more than 10,000 words. This is too long for the presentation of a few balloon profiles and model simulations that show aerosol impacts. The information in Tables 1 and 2 shows percentage disagreements between simulations and observations. This is unnecessary and corresponding figures that show model/data comparisons are sufficient."

→ We agree that some parts of the manuscript can be shortened as suggested by the reviewer. Some sections have been reorganized. However note that some details have been added at some specific locations in the manuscript as required by reviewer 2.

Firstly, the information provided in (now former) section 3.1 about the robustness of the transport calculation before investigating photochemical issues and heterogeneous processes has been reduced: the 2 first sentences and some associated references (dealing with simulation of N2O and NOy as a test for correct simulation of transport) have been removed. For consistency, part of the discussion about transport issues has been moved to section 2.2 because it actually deals with the model description. Also as a matter of consistency, the second half of (now former) section 3.1 and explaining how the in situ profile of NOy* is obtained has been moved to the discussion about 1D calculations.

Secondly, the general description of the photochemical polar summer conditions (first paragraph in former section 3.2 which is now section 3.1) is not very useful and has been removed since it focuses on the period (∼May-July) prior to the balloon cam-

Interactive
comment

paign (August-September) on which is based our study. The text in former section 3.2 (now section 3.1) has been changed to: "N2O5 is produced mainly at night from the recombination of NO2 with NO3 and destroyed during the day by photolysis leading to the reformation of NO2. Polar summer is characterised by continuous solar illumination preventing the formation of N2O5 (Fahey and Ravishankara, 1999) until about the beginning of August (Brühl et al., 1998), i.e. around day 213 for the considered Esrange/Kiruna location as illustrated in Figure 3 at 17.5 km... This situation implies that the balloon flights performed from August 7, 2009 in the Kiruna region match the photochemical conditions for which volcanic aerosols likely have an impact on NOy partitioning via elevated N2O5 hydrolysis and can be suitably used to investigate heterogeneous processes."

Thirdly, the description of the model-measurement comparisons (Section 3) has been shortened from ∼5 to less than 3 pages. Former sections 3.4 (HNO3) and 3.5 (NO2/HNO3 ratio) have been merged and reduced. The description and discussion of one-dimensional model calculations have been transferred to the new section 3.2 (NO2). Discussion about the NOx saturation effect and the description of figure 8 have been simplified. All simulations are only shown for NO2 (now figure 5). As a result, the following figures have been simplified for better clarity and to lighten the associated discussions. Former figure 9 (HNO3) has been removed.

Table 1 and Table 2 have been removed.

Section 4 about impacts on halogen chemistry has been reduced too: HCl injection discussion in section 4.1 has been shortened. Some sentences have been simplified.

Some sentences have been shortened (in particular references about Pinatubo effects on ozone) in section 5.1.

→ New figures' numbering:

Former figure 1 is now figure 2 Former figure 2 is now figure 1 Former figure 3 is now

figure 7 Former figure 4 is now figure 3 Former figure 5 is now figure 4 Former figure 6 is now figure 5 Former figure 7 is now figure 6

→ Some grammatical/typo (e.g. 'in situ' instead of 'in-situ') and other minor writing errors have been corrected throughout the text.

---

## Author Comment (AC2) · 7 Jan 2017

We thank the reviewer for his/her constructive comments which help us to improve the manuscript. Please find below our point-by-point answers.

Reviewer's comment: "However, the reasoning for exploring the effect of a moderate eruption on chemical composition at Arctic latitudes in summer must be better rationalized in the introduction. Why is this important? Currently only the relevant chemical, aerosol and some dynamical processes are explained."

$\rightarrow$ We agree with the reviewer that our introduction and the abstract can be misleading. We better explain the rationale of our study in the abstract and in the introduction. Our

main point is to study the enhanced aerosol effects in mid-latitude and high temperature conditions, not to specifically focus on impacts on polar summer chemistry which typically correspond to the late spring-July period (see POLARIS campaigns and Fahey and Ravishankara, Science, 1999). Here we pay attention on the August-September period for which we explain in new section 3.1 (formerly section 3.2 in the ACPD version) that photochemical conditions closely reflect mid-latitude conditions even at 68°N. Note that is why the title does not mention "polar summer" and remains rather general. Because these recurrent moderate eruptions span different characteristics in terms of latitude and altitude of injection (directly connected to aerosol residence times), and season (temperature effects, solar illumination for photochemical issues), different associated potential impacts on stratospheric chemistry and ozone are expected. The goal of the paper is to focus on the specific extra-vortex situation outside the wintertime low temperature conditions (i.e. comparable to mid-latitude conditions). A next step would be to investigate tropical eruptions (situation for the Kelud eruption) or injections in winter with some volcanic aerosols trapped in the polar vortex (situation for the Calbuco eruption).

We have therefore modified the end of the introduction: "...In periods following major eruptions, the year-to-year variability in stratospheric ozone at northern mid-latitudes appears closely linked to dynamical changes induced by the volcanic aerosol radiative perturbation (e.g. Telford et al., 2009; Aquila et al., 2013) and to changes in chlorine partitioning (e.g. Solomon et al., 1999; Chipperfield, 1999). Effects on stratospheric chemistry are expected in periods of elevated chlorine levels from anthropogenic activities (Tie and Brasseur, 1995; Solomon et al., 1996)... Their effects depend on the amount of released SO2 and on latitudes and altitudes of injection which directly influence aerosol residence times. The season of the eruption is also important for photochemical processes which are directly connected to temperatures and solar illuminations. The goal of this paper is to show how such moderate eruptions are likely to modify the chemical balance of the northern hemisphere lower stratosphere at periods excluding wintertime/springtime halogen-activating photochemistry. We specifically fo-

cus on the eruption of the Sarychev volcano on 15 and 16 June 2009 which injected 0.9 Tg of sulfur dioxide in the lower stratosphere (Clarisse et al., 2012) resulting in enhanced sulfate aerosol loading up to 19 km, for a period of about 8 months ending before winter (Haywood et al., 2010; Kravitz et al., 2011; O'Neill et al., 2012; Jégou et al., 2013). The approach consists in analysing some key aspects of lower stratospheric chemistry and ozone loss in a context of high aerosol surface area densities and high stratospheric temperatures using balloon-borne observations conducted in August-September 2009 from Kiruna/Esrange in Sweden (67.5°N, 21.0°E) within the frame of the STRAPOLETE project. To our knowledge we show here the first high-resolution in situ observations of chemical compounds obtained within the volcanic aerosol plume of a moderate eruption. We show that at the period on which the study is focused N2O5 has reformed and the role of its hydrolysis becomes important again after the sunlit summer period justifying the use of these balloon data for the investigation of heterogeneous processes. Aerosol-constrained simulations using a 3D Chemistry Transport Model (CTM) are compared to the observations. These model calculations ignore possible dynamical effects induced by the volcanic aerosols but are used to estimate the amplitude of the chemical impacts and ozone loss with some comparisons with the post-Pinatubo eruption period."

We have also added in the last paragraph in new section 3.1 (formerly section 3.2 in the ACPD version): "This situation implies that the balloon flights performed from August 7, 2009 in the Kiruna region match the photochemical conditions for which volcanic aerosols likely have an impact on NOy partitioning via elevated N2O5 hydrolysis and can be suitably used to investigate heterogeneous processes."

Reviewer's comment: "From the satellite data presented in Haywood (2010) and Jegou (2013) and also Fig.5 it seems obvious that the plume is not homogeneously mixed over the Arctic region by August/September 2009. Therefore the expected horizontal and vertical structure has to be discussed in some more detail. The STAC balloon data from different flights needs to be introduced in Fig. 1 (e.g. grey underlayed traces?),

not just ranges of the observations so the reader gets a better impression of the vertical structure of the plume and its variability. Horiz. and vertical variability should be discussed at least. Currently no filtering for high/low or even background aerosol regions is done for the interpretation of the data which may well be warranted but must be better supported."

→ If we have well understood the reviewer's point, our description about the mixing state of the Sarychev plume in section 2.2 (model calculations) is not consistent with results reported in the literature. It is true that space-borne data shown by Haywood et al. (2010) and Jégou et al. (2013) reflect still unmixed conditions throughout August-September 2009. In this case our discussion about the Sarychev aerosol spatial distribution in section 2.2 is not convincing and even contradictory with respect to the results presented in Figure 5 (now Figure 4 in the revised version; the new figure numbering is provided at the end of this reply). Actually, the two simulations driven by the STAC balloon-borne aerosol observations have been performed to account for the aerosol spatial inhomogeneity. In other words the two cases correspond to the 1-sigma spread of the in situ SAD observations by the STAC instrument reflecting the range of variability of the aerosol content over the northern hemisphere.

Figure 1 (which is now figure 2) has been modified accordingly. It now includes the various STAC data (average between balloon ascent and descent) and the two profiles inferred from the 1-sigma of the mean used for the Bal-sim simulation. The STAC profiles have been interpolated to a fixed 500-m vertical scale and smoothed (over 3 points, i.e. less than the plot presented on the first version of the manuscript). We have excluded data from two flights (2nd and 26th August) for which balloon outgassing is suspected as deduced from joint water vapour measurements. Also, flights revealing the sporadic presence of clouds are not considered to derive the range of SAD below 12 km.

As a result, the text in section 2.2 has been changed to: "We have conducted another type of simulation (hereafter called Bal-sim) consisting in adjusting the input $H_2SO_4$

mixing ratios so that the model output matches SADs observed by the STAC aerosol counter. Although similar aerosol SAD values were observed by Kravitz et al. (2011) in November 2009, i.e. ∼2 months after the STAC measurements as mentioned by Jégou et al. (2013), a single vertical profile may be not representative of the geographical distribution of the still unmixed volcanic plume throughout summer 2009. To account for the range of aerosol variability as observed by STAC over the Arctic region for the August-September period (Figure 2) we have performed two simulations based on the spread (1$\sigma$ standard deviation) of observed SADs. We have excluded data suspected to be spoiled by balloon outgassing as deduced from joint water vapour measurements. Also, flights revealing the sporadic presence of clouds are not considered to derive the range of SADs below 12 km. Each Bal-sim simulation is respectively driven by the lower and the upper bound of observed SAD values below 20 km (Figure 2) from the beginning of August until the end of the model run for latitudes above 40°N. Note that in Bal-sim, H2SO4 mixing ratios in July are taken from the Sat-sim simulation."

We have added some discussion about the non-homogeneity of the plume when describing Figure 5 (now figure 4): "For the Sarychev situation, minima in NO2 concentrations appear closely correlated with enhancements in aerosol amounts in the lower stratosphere (Figure 4). Thus the empirical evidence supports the view that NOx chemistry is largely driven by heterogeneous processes even in the case of a moderate volcanic eruption. The vertical structures depicted in figure 4confirm that the plume is not homogeneously mixed over the Arctic region ∼2 months after the eruption. Minimum concentration values of 1 to 2 particles.cm-3 (for sizes > 0.4 $\mu$m) correspond to unperturbed background extra-vortex conditions (Renard et al., 2010) and therefore indicate air masses unaffected by the volcanic aerosols. Conversely, layers with aerosol concentration increases by more than a factor of 3 (with respect to the mean profiles) can be assigned to the presence of the volcanic plume and show associated reductions in NO2 by up to a factor of ∼2."

Reviewer's comment: "In order to explore the sensitivities of the model study to different parameters such as differences in aerosol surface area or dynamical effects a number of differently constrained simulations have been carried out and are intercompared in the figures and tables presented. Therefore partly the figures and tables and consequently the discussion gets quite busy and confusing. The results for the runs termed sat-sim and bal-sim generally don't differ by more than 10%, mostly much less. Therefore this just needs to be shown in one plot (Fig. 6) but then can be neglected just stating that differences for other species are also minor. The same is true for the 1D simulations which are meant to check on dynamical influences. Once the results of these model experiments have been stated the following discussions can be simplified a lot by leaving all the other simulations out. Especially Table2 should be considerably simplified, I'm not aware that all the various differences presented there are even discussed in the text."

→ We have simplified the figures and shortened the text accordingly. Figure 9 of the ACPD version presenting the HNO3 profiles has been removed since it is difficult to distinguish between the various simulations which appear very close to the observation. We have written in the new section 3.3: "All the REPROBUS simulated profiles for HNO3 are mostly within the errors bars of the SPIRALE measurements and only differ by less than 10% on average (not shown). Calculated amounts from Bal-sim are increased by 10-13% when including volcanic aerosols below 19 km, highlighting limited effects on HNO3"

All the simulations, i.e. Bal-sim, Sat-sim and 1D calculations, are shown only for NO2 (now figure 5). For the NO2/HNO3 ratio we show only the Bal-sim results. The Sat-sim profile is maintained for the remote-sensing profiles of NO2 and BrO because both figures are still understandable.

The description of the model-measurement comparisons have been shortened. Sections 3.2 to (previously) 3.6 have been reduced from 3 to 2 pages. Section 3.4 (HNO3) and 3.5 (NO2/HNO3 ratio) have been merged and reduced. The description and discussion of one-dimensional model calculations have been transferred to the new sec-

tion 3.2 (NO2).

Tables 1 and 2 have been removed.

Reviewer's comment: "The balloon-borne measurements aquired during the Strapolete experiment represents a data set that nicely covers an interesting episode of aerosol enhancement in this atmospheric domain and therefore publication is of it's own value. Possibly a link or links to the appropriate data base(s) should be also given in order to enable use of the data for other studies."

→ We thank the reviewer for his/her comment. Data can be found at http://www.pole-ether.fr. This address has been added in section 2.1 of the manuscript.

Answers to detailed and minor Comments:

Reviewer's comment: "p4:l16 Is the wording "We focus here on ..." meant to discriminate against the aerosol measurements or does it just refer to gas-phase data?"

→ It refers to gas-phase data. We now write: "Our study presents in situ vertical profiles of the N2O, NO2 and HNO3 gases as observed by the SPIRALE. . ."

Reviewer's comment: "p4:l16ff Why are there only ascent data used for the SPIRALE measurements? It would be interesting to see also descent data to get a feeling on variability and possible contamination issues on ascent since the cell extends below the payload."

→ We would agree with the reviewer but firstly we have been advised to shorten the manuscript and secondly conclusions through model-SPIRALE comparisons are the same when focusing on the descent data. Thirdly, though it is always tricky to derive definitive conclusions (i.e. discriminating between contamination and fine scale variability), we have of course compared ascent/descent profiles and no obvious contamination effect, at least on a scale of 1 to a few km on the vertical axis, resulting from gondola/balloon outgassing has been pointed out. Note that independent water vapour observations were not available during the SPIRALE flights, which could have been an

interesting way to infer possible fine scale contamination effects. To summarize, contamination effects, unlikely in this case, would not change our conclusions.

Reviewer's comment: "p5:l3ff This paragraph is somewhat chaotic and hard to understand and should be polished. Before switching to the BrO profiles a new paragraph might be started."

→ We have modified the text accordingly: "Variations of solar zenith angle (SZA) along solar occultation lines of sight and associated concentration variations are likely to impact the retrieved vertical profiles near sunrise and sunset especially below 20 km (Newchurch et al., 1996; Ferlemann et al., 1998). Some works propose to use a photochemical model to correct for this effect (e.g. Harder et al., 2000; Butz et al., 2006) depending on the considered chemical compound, the observation geometry (i.e. balloon ascent or occultation) and daytime (SZA variation). Typically concentrations are converted to values expected at $90°$ SZA. In our study, the NO2 profile from the SALOMON instrument recorded on 25 August 2009 from a typical solar occultation at constant float altitude is not photochemically corrected since conversion to $90°$ SZA conditions results in differences of less than 6%, in agreement with the work of Payan et al. (1999). The vertical profile observed by the DOAS instrument was recorded on 7 September 2009 with a different observation geometry, i.e. during the balloon ascent. In this case applying a photochemical correction gives differences of 24% and the model-measurement comparison is done for SZA = $90°$."

The BrO paragraph is now separated.

Reviewer's comment: "p5:l21ff The fact that REPROBUS is used without any detailed sulfur chemistry should be clearly stated right away then referring in which different ways the aerosol plume is prescribed."

→ The REPROBUS CTM does not compute gaseous sulfur chemistry at all. Aerosols in the model are produced from H2SO4 mixing ratios provided by the UPMC 2D model (see Weisenstein and Bekki, SPARC aerosol report, chapter 6, 2006). This now specified in the Appendix and in section 2.2 (Model description). We have added in the Appendix: "Gaseous sulfur chemistry is not included in the REPROBUS CTM. The UPMC 2D model climatology (Bekki and Pyle, 1994) provides the initialization of H2SO4 mixing ratios for the background aerosols." And in section 2.2: "As sulfur chemistry is not included in REPROBUS we have conducted a simulation (hereafter called Ref-sim) constrained with typical background aerosol levels inferred from H2SO4 mixing ratios provided by the UPMC 2D model (Bekki and Pyle, 1994)."

Reviewer's comment: "p5:l53 The Haywood (2010) reference is missing."

→ Reference added.

Reviewer's comment: "p6:l33 The analytical expression for the derived correlation in Fig.3 should be given."

→ We have only plotted N2O vs NOy from MIPAS/Envisat observations in figure 3 (now figure 7 in the revised version). Conversely to the Michelsen et al.'s work we did not calculate an analytical expression for the N2O-NOy correlation since we have found unnecessary to do so in our case. The method is very simple. We have inferred the MIPAS N2O/NOy ratios for a range of (mean) stratospheric pressures, then we have interpolated these ratios to the pressure range observed during the SPIRALE flight and we have finally deduced SPIRALE "pseudo-NOy" (called here NOy*) by multiplying SPIRALE observed N2O by the interpolated ratio. We have written in the new section 3.2.2: "...An example of the estimated vertical profile of NOy (hereafter NOy*) derived from the conversion of the SPIRALE N2O profile (Figure 1a) using the N2O-NOy ratios derived from MIPAS data is presented in Figure 1b..."

Reviewer's comment: "p6:l37 Better leave out the phrase in the model."

→ The sentence has been removed.

Reviewer's comment: "p9:Sect.3.4 can be considerably shortened since it doesn't add new results (see Fig.9)."

→ As said above, former section 3.4 dealing with HNO3 has been merged with section 3.5 (NO2/HNO3 ratio) and the associated discussion substantially shortened.

Reviewer's comment: "p9:l53 Why is the nomenclature changed here to Balloonaerosim instead of Bal-sim etc.?"

→ Sorry, this was a mistake now corrected.

Reviewer's comment: "p10:Sect.3.6 The stated improvements of the 1D simulations above 20km are not at all obvious to me and are certainly not significant improvements that can be employed for the conclusion drawn in this section."

→ We agree that our initial discussion about the effect of 1D calculations was overstated. We now write: "As a result of the NOy* input in the calculations, the 1D reference simulations show very good agreement for NO2 (in red in Figure 5) with the SPIRALE measurements above 20 km. The 1D simulations constrained by observed volcanic aerosol quantities (in yellow in Figure 5) match well with the in situ measurements. The calculated chemical impact on NO2 gives percentage values similar to the 3D simulation results certainly because both NOy* and 3D NOy profiles agree well in the lower stratosphere (Figure 1b). We note that fine structures in the measured profile are not reproduced by the 1D model as a matter of height resolution and interpolation (Berthet et al., 2006). Overall the 1D NOy-constrained simulations do not significantly improve the comparisons. This result confirms that the model-observations differences in the lower stratosphere can be mostly attributed to heterogeneous processes and not to spurious calculations of transport."

Note that we now include in this new section 3.2.2 (dealing with 1D simulations) the description of the method to derive NOy* in situ profile use to drive the 1D simulation. This is more consistent than splitting the discussion about transport calculation issues in former 'Impact of transport on simulated N2O and NOy' section 3.1.

Reviewer's comment: "p12:l10ff With the introduction of Fig. 11 the dramatic difference

of the two BrO profiles should be explained (sza?). Also what is the tropopause height for the DOAS measurement? More than 1 ppt of BrO below the tropopause seems quite suspicious to me."

→ We now mention in section 4.2.1: "Differences between both profiles in terms of BrO amounts are mainly due to differences in SZA." Firstly, measured BrO at the local tropopause (here 10 km on Sept. 7, 2009 based on PTU sounding) is in agreement with previous measurements at high latitudes (Harder et al.., 1998; Dorf et al., 2006). The 2-km vertical segmentation in the BrO profile retrieval encompasses layers scanned below (upper troposphere) and above (lower stratosphere) the 10 km altitude level at which the BrO mixing ratio is provided. As a result the value at 10 km does not correspond to be purely tropospheric conditions. Secondly, the error bar at 10 km is almost compatible with 0 pptv and largely encompasses the modelled BrO. Finally, we must keep in mind that we are dealing with a lower stratosphere impacted by volcanic aerosols which are likely to enhance BrO amounts.

Reviewer's comment: "p12:l35 The meaning of the percentage changes for switching off the BrONO2 hydrolysis must be more clearly explained. For the example given it should be 16% of the daytime BrO production not 18%."

→ Sorry, this was a mistake. It is indeed 16% of BrO production due to BrONO2 hydrolysis. We have rewritten the sentence: "It particularly shows that under the Sarychev aerosol loading, only 16% of the 22% (0.9 pptv) increase in daytime BrO at 16.5 km for the August-September 2009 period is produced from BrONO2 hydrolysis."

Reviewer's comment: "p12:l39 This result implies ..."

→ corrected

Reviewer's comment: "p12:l52 ... the active chlorine family species ..."

→ corrected

Reviewer's comment: "p13:l30ff I propose to use absolute values for the ozone loss

discussion (see the comment on Fig.12)."

Reviewer's comment: "p13:l39 ... into the lower stratosphere."

→ This sentence has been changed following reviewer1's remark.

Reviewer's comment: "p15:l33 When switching to absolute values for the accumulated ozone losses the discussion why "largest" losses occur just above the tropopause will become obsolete, I guess."

→ Indeed, the sentence has been removed.

Reviewer's comment: "p16:l35 It might be interesting ..."

→ corrected

Reviewer's comment: "p18:l19 Does the uncertainty represent accuracy? Since several balloon-borne measurements are used along with each other and are compared to other aerosol SAD data it is not sufficient to just state the precision of the measurements."

→ We are not sure to understand the reviewer's point. Here (p18 l19) we focus on aerosol counting observations with 3 copies of the STAC instrument which have shown $\pm10\%$ differences (random error or precision) using identical aerosol at high concentrations in the laboratory. Poisson statistics provide an estimate of the random error for lower aerosol concentration conditions (i.e. 60% for aerosol concentrations of $10-3$ cm$-3$, 20% for $10-2$ cm$-3$, and 6% for concentrations higher than $10-1$ cm$-3$). This calculation of aerosol counting uncertainties (precision) is detailed in Deshler et al. (2003). This reference is provided in this part of the manuscript. The accuracy (or measure of the bias if the reviewer is dealing with this other definition) of this type of instrument is very difficult to derive because one would need reference concentration values (true value from another aerosol counter) which are extremely difficult to obtain. In other words, no absolute reference aerosol counter is available. That is why, as in Deshler et al. (2003) we provide here precision. We have then changed the text to:

"Using a statistical approach as described in Deshler et al. (2003), STAC counting precision (Poisson statistics and the ±10% measurement reproducibility) translate into uncertainties on distribution moments, with estimated values of 40% for SAD."

Reviewer's comment: "p18:l41 I guess overall uncertainty represents accuracy? Rename or otherwise state the accuracy."

→ This is total error combining precision and accuracy. "Overall uncertainties" has been replaced by "total error".

Reviewer's comment: "p19:l17 The Voigt et al. reference is missing. Also the Pfeilsticker et al. reference (l20). Please check over completely!"

→ We have checked all the references and added the missing ones.

Reviewer's comment: "p21:l3 ... strong functions ..."

→ corrected

Reviewer's comment: "p22 All citations should be thoroughly rechecked since several citations have been missing from the references list."

→ We have checked all the references and added the missing ones.

Reviewer's comment: "Table 2 Several of the tabulated differences are not used at all in the text. The table can be reduced considerably or even removed completely."

→ Table 1 and 2 have been removed.

Reviewer's comment: "Table 3 In the caption it should better read: Numbers are taken from the Sat-sim simulation. Also ... BrONO2 hydrolysis (Reaction 4) to changes ... would help."

→ Changes done in the caption.

Reviewer's comment: "Fig.1 I propose to show the traces of the individual measurements on the plot to give the reader an impression on the variability. Also the altitudes

shown should be extended somewhat to 25km and possibly below 10km to give a better impression on the vertical extend of the plume."

→ As stated above Figure 1 (now ranked as figure 2) has been modified (almost) accordingly. We have added the individual profiles (excluding suspected balloon/gondola outgassing cases) of Surface Area Densities (SAD) inferred from STAC size distributions (using a log-normal fitting procedure as described in Jégou et al., 2013) but this calculation was not done for altitudes above ∼20 km (model outputs are used). The altitude range of the plume is still visible on the 10-20 km range. Below 10 km the presence of tropospheric clouds (depending on the varying tropopause height) is likely to spoil the interpretation of the SAD profiles. That is why some profiles have been truncated on the new figure 2. Note that the whole aerosol concentration profiles can be found in Renard et al. (2010).

Reviewer's comment: "Fig.2 For my taste in the right panel SPIRALE HNO3 should be included."

→ The idea of the reviewer is interesting but the figure (now figure 1b) would become very busy (this is already the case here) when including HNO3 and therefore rather confusing for the reader. That is why we have decided to keep the figure as is.

Reviewer's comment: "Fig.5 The grey shaded (error bars?) on the NO2 profile should be explained in the caption."

→ These are indeed error bars. This is now indicated in the caption.

Reviewer's comment: "Fig.9 Due to the high partitioning into HNO3 this species is not sensitive for the aerosol effect. This plot therefore can be left out."

→ The plot has been removed as said above.

Reviewer's comment: "Fig.11 The solar zenith angles of the profiles have to be given in the caption due to the diurnal variation of BrO."

→ The following information is now provided in the caption: "The SALOMON data in the lower stratosphere were obtained between 19:15 UT (SZA=93.8° at 22 km tangent height) and 19:25 UT (SZA=94.5° at 17 km tangent height). The DOAS profile was measured between 15:15 UT (SZA=77.5° at 10 km) and 15:55 UT (SZA=81.3° at 22 km) during balloon ascent."

Reviewer's comment: "Fig.12 Showing the percentage changes in ozone is somewhat misleading. In order to point out the altitude regime of highest impact in terms of ozone loss an absolute scale should be used like loss rates of ppb/day. A percentage change of 5% doesn't have any major effect if ozone levels are negligible at the TP."

→ The point highlighted by the reviewer is relevant. Most papers referring to the Pinatubo aerosol-induced ozone loss provide plots with % values when the loss is expressed vs altitude levels (e.g. Brasseur and Granier, 1992; Pitari and Rizi, 1993; McGee et al., GRL, 1994; Kinnison et al., 1994; Tie et al., 1994; Solomon et al., 1996). That is the reason why chose to plot percentage values.

To address the reviewer's comment, we now plot (cumulated) ozone loss in pbbv (as is widely expressed in the literature regarding polar ozone depletion, not really in ppbv/day which in the case of the Sarychev will give too low numbers). We still provide in the text associated percentage values for comparison with the literature dealing with Pinatubo effects.

This is provided in section 5.1: "Accumulated ozone depletion reaches its maximum above Kiruna near 16 km from around mid-September with changes of -22 ppbv corresponding to -1.5%. Below this level changes range from -10 ppbv to -18 ppbv, i.e. -2.5% to -3.5%. From the upper bound of the Bal-sim outputs calculated ozone depletion reaches -25 ppbv (-2.8%) and -35 ppbv (-4%) at 16.5 km and 14 km, respectively (not shown). It should be noted that at the tropopause level the possible detailed influence of various organic compounds originating from the upper troposphere is not taken into account in our simulations. We note that for the post-Pinatubo eruption period, ozone reductions as large as -30% were measured for the 12 and 22 km altitude range monitored at some mid-latitude locations in winter and spring (Hofmann et al., 1994) but these losses are both due dynamical and chemical perturbations. Through 2D modelling, ozone losses of up to -20% directly resulting from heterogeneous chemical processes were calculated in the northern hemisphere lower stratosphere over the first year following the Pinatubo eruption (Pitari and Rizi, 1993; Tie et al., 1994). The calculated chemical loss had reduced to values much closer to those simulated for the Sarychev aerosols, i.e. ∼-5%, at 60°N in the autumn 1992 extra-polar vortex conditions (Tie et al., 1994)."

We have added in the abstract: "As a consequence, the simulated ozone loss due to the Sarychev aerosols is low with a reduction of -22 ppbv (-1.5%) of the ozone budget around 16 km".

And added in the conclusion: "...the magnitude of the ozone response to the Sarychev volcanic perturbation appears restricted for instance -22 ppbv or -1.5% at 16 km)..."

And we have to remove the following sentence in the conclusion (p15 line 34 in the ACPD version): "Eventually, the largest ozone destruction is restricted to the lowermost stratosphere (the bottom of the volcanic aerosol layer close to the tropopause) where catalytic cycles are primarily controlled by HOx and where the NOx photochemistry plays a very minor role."

→ It is important to note the new figures' numbering:

Former figure 1 is now figure 2 Former figure 2 is now figure 1 Former figure 3 is now figure 7 Former figure 4 is now figure 3 Former figure 5 is now figure 4 Former figure 6 is now figure 5 Former figure 7 is now figure 6